

# Phosphorus cycling in the upper waters of the Mediterranean Sea (Peacetime cruise): relative contribution of external and internal sources

Elvira Pulido-Villena[1], Karine Desboeufs[2], Kahina Djaoudi[1a], France Van Wambeke[1], Stéphanie Barrillon[1],

Andrea Doglioli[1], Anne Petrenko[1], Vincent Taillandier[3], Franck Fu[1b], Tiphanie Gaillard[1], Sophie Guasco[1],

Sandra Nunige[1], Sylvain Triquet[2], Cécile Guieu[3]

[1]Aix-Marseille Université, CNRS/INSU, Université de Toulon, IRD, Mediterranean Institute of Oceanography (MIO) UM 110, 13288, Marseille, France

[2]LISA , Université de Paris, Univ. Paris-Est Créteil, CNRS, UMR7583, Créteil, France

[3]CNRS, Sorbonne Université, Laboratoire d'Océanographie de Villefranche (LOV), UMR7093, 06230 Villefranche-sur-Mer, France

[a]Now at: University of Arizona, department of Molecular and Cellular Biology, 1007 E Lowell Street Life Science South, room 315 Tucson, AZ 85721, USA

[b]Now at: The W. M. Keck Science Department of Claremont McKenna, Scripps, and Pitzer Colleges, Claremont, CA 91711,

USA

*Correspondance to*: Elvira Pulido-Villena (elvira.pulido@mio.osupytheas.fr)

## Abstract

The study of phosphorus cycling in P-depleted oceanic regions, such as the Mediterranean Sea, has long suffered from methodological limitations leading to a simplistic view of a homogeneous surface phosphate pool with concentrations

theoretically set to zero above the phosphacline. During the PEACETIME (Process studies at the air-sea interface after dust deposition in the Mediterranean Sea) cruise, carried out from 10 May to 11 June 2017, we conducted collocated measurements of phosphate pools at the nanomolar level, alkaline phosphatase activities and atmospheric deposition of phosphorus, across a longitudinal gradient from the west to central Mediterranean Sea. In the phosphate depleted layer (PDL), between the surface and the phosphacline, nanomolar phosphate was low and showed little variability across the transect spanning from $6 \pm 1$ nmol L-1 in

the Ionian basin to $15 \pm 4$ nmol L-1 in the westernmost station. The low variability in phosphate concentration contrasted with that of alkaline phosphatase activity which varied over one order of magnitude across the transect. Nanomolar phosphate data revealed density gradients of phosphate concentration inside the PDL ranging between $10.6 \pm 2.2$ µmol kg-1 in the westernmost station to values close to zero towards the east. Using the density gradients, we estimated diapycnal fluxes of phosphate to the PDL and compared them to atmospheric deposition, another external source of phosphate to the PDL. Phosphate supply to the

PDL from dry deposition and diapycnal fluxes was comparable in the western part of the transect. The contribution of atmospheric deposition to external P supply increased under the occurrence of rain and Saharan dust. This result contrasts with the longtime idea that, under stratification conditions, the upper waters of the Mediterranean Sea receive new P mainly exclusively from the atmosphere. Although this finding must be taken cautiously given the uncertainties in the estimation of diapycnal fluxes, it opens exciting questions on the biogeochemical response of the Mediterranean Sea, and more generally of

marine oligotrophic regions, to expected changes in atmospheric inputs and stratification regimes. Taken together, external sources of phosphate to the PDL contributed little to total phosphate requirements which were mainly sustained by in situ hydrolysis of DOP. The results obtained in this study show a highly dynamic phosphorus pool in the upper layer of the euphotic





zone, above the phosphacline, and highlight the convenience of combining highly sensitive measurements and high-resolution

sampling to precisely depict the shape of phosphate profiles in the euphotic zone with still unexplored consequences on P fluxes

supplying this crucial layer for biogeochemical cycles.

## 1 Introduction

In the oligotrophic ocean, which covers > 60% of the global ocean, biological activity and carbon export are constrained by

nutrient availability. By controlling the efficiency of the biological pump, N and P determine the strength of the oceanic carbon

uptake (Falkowski et al., 1998, Moore et al., 2013). Concentration of dissolved nutrients are kept low by microorganisms within

the euphotic zone and their replenishment is partly driven by the physical structuration of the water column (Lewis et al., 1998).

Vertical diapycnal fluxes supply nutrients from the ventilated nutrient-rich deep layers to the base of the euphotic layer

sustaining production in sub-surface waters, generally just below the pycnocline. However, the upper euphotic zone, above the

pycnocline, has been traditionally seen as isolated from this nutrient supply from below. Consequently, vertical profiles of

nutrients typically exhibit two layers: a so-called nutrient depleted layer in which nutrient concentration approaches zero,

spanning between the surface and the top of the nutricline; and a nutrient repleted layer, across the nutricline to the bottom.

Under the effects of global warming, stratification regimes may be globally modified mainly through enhanced density gradients

at the base of the mixed layer depth. An increase in upper ocean stratification has recently been reported for the last half-century

(Li et al., 2020) and although most models predict further increases at the global scale (Capotondi et al., 2012), regional

differences may arise (Macias et al., 2015, Somavilla et al., 2017). In this context, it is crucial to accurately understand nutrient

cycling in the surface oligotrophic ocean and to accurately quantify nutrient sources sustaining biological productivity and their

evolution.

Phosphate plays a central role in the biogeochemical function of the ocean (Karl, 2014) and there is a growing awareness of its

deficiency in certain oligotrophic regions in the North Atlantic Ocean and the North Pacific Ocean (Ammerman et al., 2003;

Letelier et al., 2019; Martiny et al., 2019). In oceanic regions not limited by iron, P is commonly regarded as either the ultimate

limiting nutrient, as any N deficit may be offset by $N_2$ fixation (e.g. Moutin et al., 2008; 2018), or as a proximate limiting

nutrient in regions such as the North Atlantic sub-tropical gyre (Wu et al., 2000) and the Mediterranean Sea (Mermex group,

2011). The study of phosphate cycling in P-depleted oceanic regions has long suffered from methodological limitations leading

to a common view of a homogeneous surface phosphate pool with concentrations theoretically set to zero above the

phosphacline. With the advent of sensitive techniques of analysis, this black box is being opened revealing high spatial

variability of surface phosphate concentration at nanomolar levels at both regional (Wu et al., 2000; Djaoudi et al., 2018a) and

global scale (Martiny et al., 2019). The increasing analytical accuracy has also allowed, for instance, revealing changes in

phosphate concentration in response to external forcing such as dust deposition (e.g. Pulido-Villena et al., 2010, Mahaffey et al.,

2014) or characterizing the role of phosphate as a driver of ecosystem functioning (Martiny et al., 2019).

In P-deficient regions, dissolved organic phosphorus (DOP) constitutes an alternative source of phosphate for microorganisms

(Karl, 2014). Most microorganisms synthesize one or more phosphohydrolytic enzymes in order to degrade selected DOP

compounds for which the most commonly studied is alkaline phosphatase (AP). High AP activities have been typically reported

in subtropical gyres of the North Atlantic (Mather et al., 2008; Lomas et al., 2010) and North Pacific (Duhamel et al., 2011)

Oceans as well as in the Mediterranean Sea (Sala et al., 2001, Van Wambeke et al., 2002) suggesting an active utilization of DOP

by microorganisms when phosphate is scarce. However, the methodological limitations mentioned above have translated into



insufficient concomitant measurements of AP activity and phosphate pools in the upper ocean (Hashihama et al., 2020)
      precluding an empirical assessment on the interplay between phosphate concentration and microbial P cycling.

      The Mediterranean Sea shows a long summer stratification period that translate into relatively low-nutrient concentrations and an
      overall phosphorus (P) deficit compared to nitrogen (N) evidenced by higher N:P than Redfield ratios of inorganic nutrients in
      deep layers (Krom et al., 2010, Pujo-Pay et al., 2011). These patterns are accentuated towards the east, translating into a marked

longitudinal gradient of oligotrophy (Mermex Group, 2011). Mainly based on N-excess relative to Redfield ratio, the
      Mediterranean Sea has been generally considered as a P-limited oceanic region (Krom et al., 2005; 2010). Actually,
      phytoplankton experiences N and P (co)-limitation (Tanaka et al., 2011) and heterotrophic prokaryotes are usually P-limited or
      co-limited by P and labile C (Van Wambeke et al., 2002). Recently, the use of high-sensitive phosphate measurements has
      revealed surface phosphate concentrations as low as 1 nM (Pulido-Villena et al., 2010; Djaoudi et al., 2018a) and high AP

activity rates suggest an active microbial cycling of P (Van Wambeke et al., 2002).

      The Mediterranean Sea continuously receives anthropogenic aerosols originating from industrial and domestic activities from all
      around the basin and other parts of Europe as well as a noticeable flux of dust from the Sahara (Mermex Group, 2011). Both
      anthropogenic and natural atmospheric deposition constitute a key external source of dissolved macronutrients such as P
      (Ridame and Guieu, 2002; Pulido-Villena et al., 2010) and N (Markaki et al., 2003) to Mediterranean surface waters. Climate

change effects are predicted to play a prominent role in modifying the biogeochemical functioning of the Mediterranean Sea
      (Richon et al., 2019). Modifications in atmospheric deposition fluxes (Kanakidou et al., 2020) and in stratification regimes
      (Macias et al., 2015) will likely impact phosphate supply to the mixed layer with presumed consequences on microbial P cycling
      and productivity.

      Here we present the first cross-basin simultaneous measurements of phosphate concentration at nanomolar level and alkaline

phosphatase activities in the P-depleted upper waters of the Mediterranean Sea during the season characterized by strong
      stratification. We first explore the longitudinal trends of surface phosphorus cycling. We then estimate the contribution of
      vertical diffusion to the total supply of new P to the mixed layer and we compare this source from below with phosphate fluxes
      from the atmosphere and from enzymatic hydrolysis of DOP.

## 2 Material and methods

### 2.1 Cruise overview and seawater sampling


      This study was conducted as part of the PEACETIME cruise (doi.org/10.17600/15000900) on board the R/V 'Pourquoi Pas?' in
      the Mediterranean Sea, from May 10[th] to June 11[th], 2017. A total of 13 stations were visited along a transect extending from the
      western basin to the center of the Ionian Sea (25°S 115 E –15°S, 149°W, Fig. 1). For details on the cruise strategy, see Guieu et
      al. (2020). For this study, 10 stations located across a longitudinal gradient were selected: ST4, ST9, ST10, and FAST were

located in the Algerian basin, ST5, TYR and ST6 in the Tyrrhenian basin and ST7, ST8 and ION in the Ionian basin (Fig. 1). All
      stations were sampled with both classical CTD-rosette and TMC-rosette (trace metal clean conditions). Stations FAST, TYR and
      ION were long duration stations and they were sampled several times during at least 4 days, alternating CTD and TMC casts. For
      samples collected at these three long stations, only the first deployed CTD- and TMC-rosette casts were considered for this
      study. The long stations were sampled at high vertical resolution inside the 0-200 m layer.





Seawater samples for dissolved phosphorus analyses were collected from the TMC rosette between the surface and the bottom and filtered online through 0.2 µm (Sartorius Sartobran P®) into 60 mL HDPE-bottles previously cleaned with hydrochloric acid (HCL) and ultrapure water. Seawater samples for alkaline phosphatase activity and particulate organic phosphorus (POP) were collected from the CTD rosette at 5 m depth.

**2.2 Dissolved and particulate phosphorus analysis**

Dissolved inorganic phosphate (hereafter 'phosphate') was analyzed onboard immediately after sampling using a segmented flow analyzer (AAIII HR Seal Analytical) according to Aminot and Kérouel (2007) with a limit of quantification of 0.02 µmol L$^{-1}$. Samples with concentration below the limit of quantification were analyzed using the Liquid Waveguide Capillary Cell method (LWCC) according to Pulido-Villena et al. (2010). The LWCC was 2.5 m long and the limit of detection was 1 nmol L$^{-1}$. The analytical precision, estimated through repeated measurements of a surface seawater sample was 4.5% (n= 10).

Samples for dissolved organic phosphorus (DOP) were stored frozen until analysis back in the laboratory. After the cruise, samples for DOP were thawed at ambient temperature in the laboratory. Total dissolved phosphorus (TDP) was analyzed using the LWCC technique after UV-digestion (Armstrong et al., 1966) during 3 hours using a Metrohm® 705 UV digester. The efficiency of digestion was assessed using two 100 nM solutions of glycerol-phosphate and glucose-phosphate and was 99 ± 1% (n = 6) and 100 ± 3% (n = 3), respectively. DOP was obtained by the difference between TDP and nanomolar phosphate.

Samples for particulate organic phosphorus (POP, 1.2 L) were filtered onboard on pre-combusted (4h, 450°C) glass fiber filters (Whatman GF/F 47 mm). The filters were stored in cryotubes at -20°C until analysis after the cruise. POP was converted to phosphate using the wet oxidation method based on a persulfate digestion at 120°C (Raimbault et al., 1999) and analyzed using the segmented flow analyzer. The limit of quantification, determined by analyzing 10 blank filters, was 1 nmol L$^{-1}$.

**2.3 Alkaline phosphatase activity measurements**

Alkaline phosphatase activity (APA) was measured fluorometrically, using 4 methylumbelliferyl – phosphate (MUF-P) as fluorogenic model substrate (Hoppe, 1983). Stocks solutions (5 mM) were prepared in methylcellosolve and stored at -20°C. The release of the product of hydrolysis, MUF, was followed by measuring the increase of fluorescence (excitation/emission 365/450 nm, wavelength width 5 nm) in a VARIOSCAN LUX microplate reader. The instrument was calibrated with standards of MUF solution diluted in < 0.2 µm filtered seawater. For measurements, 2 ml of unfiltered seawater samples were spiked with 100 µl of

MUF-P solution diluted so that different concentrations (0.025 to 1 µM) were dispatched into a black 24-well polystyrene plate in duplicate. Incubations were run in the dark in thermostated incubators reproducing *in situ* temperature and lasted up to 24 h with a measurement of fluorescence every 1 to 3 h depending on expected activities. The rate was calculated from the linear part of the fluorescence versus time relationship.

The parameters V$_{max}$ (maximum hydrolysis velocity) and K$_m$ (Michaelis-Menten constant which reflects enzyme affinity for the
substrate) were determined by fitting the data using a non-linear regression on the equation:

$$V = \frac{(V_{max} \times S)}{(K_m + S)}$$ (1)

where V is the hydrolysis rate, and S the concentration of MUF-P added.

The turnover time of the added substrate, MUF-P, was estimated by the K$_m$:V$_{max}$ ratio.



### 2.4. Aerosol and rain sampling and analysis

Details on atmospheric sampling and analyses are given in three companion papers of this special issue (Van Wambeke et al. 2020, Desboeufs et al. and Fu et al. in preparation). A brief description is given hereafter.

The PEGASUS container was installed onboard the R/V Pourquoi Pas? (Guieu et al., 2020). Atmospheric aerosol sampling was performed using isokinetic and wind-oriented aerosol multi-samplers, enabling a sampling of particles with aerodynamic diameter inferior to 40 μm (Rajot et al., 2008). Aerosol particles were collected on 47-mm polycarbonate membranes of 0.4 μm pore size (Whatman Nuclepore TM) for measuring their nutrients contents. The volume flow rate was set at 20 L min$^{-1}$. All the

filters were previously cleaned by immersion in ultrapure HCl (2%) during 2 hours and rinsing with ultrapure waters. A sampling strategy was specifically set up to avoid contamination from the ship smoking. In total, 36 filters were collected which 17 filters during the stations and 19 on the route. Five blanks of filters were also prepared. Moreover, two wet deposition events were sampled on board during the cruise, one at ION and one at FAST. The rain sampling was conducted with an on-line filtration collector (0.2 μm), enabling to discriminate the dissolved and particulate fractions of phosphorus.

Soluble P concentration in aerosols was estimated after leaching of aerosol filters in ultrapure water. The dissolved samples were then analyzed by HR-ICP-MS (Neptune Plus, Thermo Scientific ™) for total soluble P concentrations. For rain, dissolved inorganic phosphate concentration was measured by ion chromatography (Metrohm, model 850 Professional IC with Metrosep A Supp 7 column for anions measurements) in the dissolved fraction of the 2 rain samples. Total dissolved phosphorus (TDP) concentration in rain was obtained by HR-ICP-MS. The atmospheric dissolved organic phosphorus (DOP) was estimated from

the difference between TDP and dissolved inorganic phosphate.

### 2.5 Data analysis and calculations

*Phosphacline and mixed layer depths.* The phosphacline was defined for each station as the layer with maximum density gradient of phosphate concentration, computed as the highest significant slope of the linear fitting of DIP concentration versus density. The intercept of the regression line, the phosphate depletion density, which is the deepest isopycnal at which phosphate

concentration is zero (Kamykowski and Zentara, 1986; Omand and Mahadevan, 2015), was used to estimate the phosphacline depth (Table 1).

The mixed layer depth (MLD) was determined, using the shape of CTD profiles, as the depth where the residual mass content is equal to 1 kg m$^{-2}$, with an estimation error 0.5 m relative to the vertical resolution of the profile (1 m).

*Vertical fluxes of phosphate.* The vertical fluxes of phosphate were estimated following Taillandier et al. (2020). In summary,

vertical fluxes are parameterized as diapycnal diffusive fluxes, written as the product of the eddy vertical diffusivity ($K_{turb}$) and the vertical gradient in phosphate concentration (C):

$$F_{PO4} = K_{turb} \cdot \partial C / \partial z \qquad (2)$$

The eddy vertical diffusivity was calculated using the Osborn's (1980) relationship with a constant mixing efficiency of 0.2:

$$K_{turb} = 0.2 \cdot \varepsilon / N_2 \qquad (3)$$

where ε is the turbulent kinetic energy dissipation rate and N is the buoyancy frequency expressed as:





$$N = - (g/\rho 0 \cdot \partial\rho/\partial z)^{1/2} \tag{4}$$

where g is the gravitational constant and $\rho_o$ the reference density of seawater.

Substituting Eq.(4) in Eq.(3) and then in Eq.(2), the vertical flux of phosphate is expressed as the product of ε and the gradient of phosphate concentration across isopycnals, obtained by linearly fitting phosphate concentrations versus density:

$$F_{PO4} = -0.2 \cdot \varepsilon \cdot \rho_0/g \cdot \partial C/\partial \rho \tag{5}$$

This formulation is considered to have a stronger temporal consistency in the density-nutrient relationship than in the depth-nutrient relationship (Omand and Mahadevan, 2015). During the PEACETIME cruise, there were no direct measurements of turbulent kinetic energy dissipation rates (ε). Therefore, to estimate at best vertical fluxes of phosphate we relied on

measurements reported by Cuypers et al. (2012) in the same geographical area (6-10 $10^{-9}$ W kg$^{-1}$ for the 20-100 m layer). The objective of this approximation was to scale the measured phosphate gradients into flux units. The uncertainties associated with such calculation will be further discussed.

*Atmospheric deposition fluxes.* Onboard aerosol characterization showed the predominance of background and anthropogenic aerosol along the cruise, except at FAST where a Saharan dust plume was sampled (Fu et al., this issue). Atmospheric dry

deposition of soluble inorganic P was estimated using the soluble P aerosol concentration multiplied by a dry settling velocity of 1 cm s$^{-1}$, typical of anthropogenic aerosol particles, and, exceptionally, of 3 cm s$^{-1}$ at the FAST as this value is more adapted for Saharan events (Izquierdo et al., 2012). At each short duration station (ST4, ST5, ST6, ST7, ST8, ST9, and ST10), soluble P flux was determined from the filter sampled during the period of the occupation of the short station, ranging 0.28 - 1.15 days. At the 3 long stations (FAST, TYR and ION), we used the aerosols filters collected during the periods comprising the CTD cast from

which the P concentrations in the water column were measured.

The wet deposition P fluxes were estimated from the measured dissolved concentrations in the rain samples multiplied by the total precipitation accumulated during the period of the rain over the R/V location (Desboeufs et al., this issue). The total precipitation during the rain events were issued from the hourly total precipitation accumulated on the grid-point spanning R/V position from ERA5 data reanalysis. The accumulated rate was 3.5 ± 1.2 mm at ION and 5.7 ± 1.4 mm at FAST.

*In situ AP hydrolysis rates.* In situ hydrolysis rates for alkaline phosphatase were computed using an estimated hydrolysable fraction of measured DOP (AP-DOP) as substrate concentration (S) in Michaelis-Menten kinetic equation as following:

$$V_{insitu} = \frac{\left(V_{max} \cdot AP - DOP\right)}{\left(K_m + AP - DOP\right)} \tag{6}$$

AP-DOP was estimated assuming a fraction of AP-DOP to DOP of 31% ± 18%, average value from published data (Djaoudi et al., 2018b) and unpublished measurements in the study area (n=36).



## 3 Results

**3.1 Biogeochemical features of the phosphorus pool**

Figure 1 shows phosphate vertical profiles between the surface to the bottom at the ten visited stations across a longitudinal gradient from 1.6°E (ST10) to 19.8°E (ION). Phosphate vertical profiles exhibited two characteristic layers in all sampled stations: a phosphate depleted layer (hereafter PDL), from the surface to the upper boundary of the phosphacline, and a phosphate repleted layer (hereafter PRL), below the phosphacline to the bottom (Fig. 1).

Phosphate concentration in the PRL, measured through the standard technique,  ranged from $0.40 \pm 0.02$ µmol L$^{-1}$ in the Algerian basin (ST10, FAST, ST9 and ST4) to $0.35 \pm 0.04$ µmol L$^{-1}$ in the Tyrrhenian basin (ST5, TYR and ST6) and $0.18 \pm 0.03$ µmol L$^{-1}$ in the Ionian basin (ST8, ST7 and ION) (Fig. 1). Maximum density gradient of phosphate concentration, indicative of the phosphacline, ranged from $356 \pm 51$ µmol kg$^{-1}$ in the Algerian basin and $396 \pm 159$ µmol kg$^{-1}$ in the Tyrrhenian basin, to $886 \pm 73$ µmol kg$^{-1}$ in the Ionian basin (Table 1). The phosphacline depth showed similar values in the Algerian and Tyrrhenian basins

(between 61 m at ST5 and 90 m in ST9) and was higher than 120 m in the Ionian basin (Table 1, Fig. 2A).

At all sampled stations, phosphate concentration in the PDL fell below the limit of detection of the standard technique, spanning from $6 \pm 1$ nmol L$^{-1}$ in ST7 in the Ionian basin to $15 \pm 4$ nmol L$^{-1}$ in the westernmost station, ST10 (Fig. 2B, Table 2). Dissolved organic phosphorus (DOP) in the PDL showed minimum values at ST5, ST8 and ION, in the central and eastern part of the transect ($30 \pm 3$ nmol L$^{-1}$, $31 \pm 2$ nmol L$^{-1}$ and $36 \pm 10$ nmol L$^{-1}$, respectively, Fig. 2C) and the highest DOP concentration was

observed at FAST (100 nM, Fig. 2C, Table 2). DOP represented between 70 and 89 % of the total dissolved phosphorus pool with no particular trend across the transect. Particulate organic phosphorus (POP) in the PDL, measured at 5 m depth, decreased towards the East and ranged between $25 \pm 2$ nM in ST5 and $11 \pm 1$ nM in ION (Fig. 2E, Table 2). POP and, to a lesser extent, phosphate and DOP concentration showed a decreasing trend towards the East (Table 3).

A full description of alkaline phosphatase (AP) rates and corresponding Michaelis-Menten kinetics is detailed  in Van Wambeke

et al. (2021). Across the longitudinal gradient (Fig. 1), AP maximum hydrolysis rates ($V_{max}$) at 5 m depth regularly increased toward the east (from $0.5 \pm 0.1$ nmol P L$^{-1}$ h$^{-1}$  to $5.6 \pm 0.2$ nmol P L$^{-1}$ h$^{-1}$ (Fig. 2E, Table 2) with a mean value of $2.1 \pm 1.6$ nmol P L$^{-1}$ h$^{-1}$. AP half-saturation constant ($K_m$) depicted no longitudinal trend ranging from  $73 \pm 21$ nM  (ST4) to $280 \pm 28$ nM (ST9) (Fig. 2F, Table 2). Turnover time of spiked DOP substrate, MUF-P, ($T_{MUF}$) ranged between 22 (ST7) and 246 hours (ST10). AP $V_{max}$ normalized by POP (AP-POP) ranged between 0.02 h$^{-1}$ in the Algerian basin (ST4) and 0.51 h$^{-1}$ in the Ionian basin (ION)

(Table 2). Both AP-POP and $T_{MUF}$ exhibited a significant longitudinal trend but with no correlation with phosphate nor DOP concentration (Table 4).

**3.2 Vertical distribution and density gradient of phosphate concentration in the phosphate depleted layer**

Nanomolar phosphate data revealed differences in the vertical homogeneity of phosphate concentration among stations inside the PDL (Fig. 2B). In the Ionian basin, phosphate concentration was rather constant between the surface and the phosphacline

(average CV 14%) and more variable in Tyrrhenian (24%) and Algerian (32%) basins. Vertical profiles showed increasing phosphate concentrations with depth from the surface to the upper bound of the phosphacline (Fig. S1). Indeed, phosphate concentration in the PDL may be linearized against density in most sampled stations of the transect (Fig. 3, Table 3, Fig. S1). The exceptions to this linear relationship were (1) ST6 and ION, where the slope of the regression line (i.e. density gradient of phosphate concentration, $\partial C/\partial \rho$) was not significantly different from zero, and (2) ST4 and ST5, where an increase of phosphate





with density was observed although it was not linear (Fig. S1). The case of the easternmost station, ION, deserves further attention. In this station, the PDL was divided in two sub-layers: a first one, between the surface and 66 m depth, where phosphate concentration was low, homogeneous and no $\partial C/\partial\rho$ was observed; and a second one, between 66 m and the top of the phosphacline (166 m), where DIP concentration was still low (below 30 nM) but a first $\partial C/\partial\rho$ of $143 \pm 26$ µmol kg$^{-1}$ appeared above the maximum gradient indicative of the phosphacline ($936 \pm 43$ µmol kg$^{-1}$) (Fig. 3C, Table 1; Table 3).

The density gradient of phosphate concentration ($\partial C/\partial\rho$) inside the PDL ranged from $10.6 \pm 2.2$ µmol kg$^{-1}$ in ST10 (Algerian basin), to undetectable values in ST6 and ION (Table 3). $\partial C/\partial\rho$ inside the PDL decreased significantly with longitude ($r = -0.71$, $p < 0.05$, $n = 10$) and was negatively correlated with phosphacline depth ($r = -0.69$, $p < 0.05$, $n = 10$). $\partial C/\partial\rho$ inside the PDL was two orders of magnitude lower than across the phosphacline (Table 1).

### 3.3 Phosphate fluxes to the phosphate depleted layer

Data obtained during the cruise allowed the estimation of three phosphate fluxes supplying the PDL: two external, diapycnal and atmospheric fluxes, and one internal through the enzymatic hydrolysis of DOP.

Using measured density gradients inside the PDL (Table 1) and reported values of turbulent kinetic energy dissipation rate ($\varepsilon$) for the Mediterranean Sea (see section 2.5), we estimated diapycnal fluxes of phosphate ($F_{PO4}$) to the PDL. They ranged between 0 in ST6 and ION (where phosphate concentration inside the PDL was homogeneous and no density gradient was observed) and 0.16

$\pm 0.09$ µmol P m$^{-2}$ d$^{-1}$ in ST4 in the Algerian basin (Fig. 4). Fluxes across the phosphacline, also computed as described in section 2.5, ranged between 4.5 and 14.4 µmol m$^{-2}$ d$^{-1}$.

Soluble phosphate fluxes derived from dry atmospheric deposition at each station (see section 2.2) were highly variable across the transect with values ranging between $0.027 \pm 0.012$ µmol m$^{-2}$ d$^{-1}$ (ST10) and $0.995 \pm 0.050$ µmol m$^{-2}$ d$^{-1}$ (FAST) during a Saharan dust event (Fu et al., this issue) (Table S1, Fig. 4). In addition, two rain events were sampled at ION and FAST

supplying dissolved phosphate wet deposition values of $0.663 \pm 0.227$ and $1.146 \pm 0.290$ µmol P m$^{-2}$ d$^{-1}$, respectively (Desboeufs et al., this issue).

*In situ* DOP hydrolysis fluxes by alkaline phosphatase computed using Michaelis-Menten kinetics as defined in section 2.5, integrated over the mixed layer, increased eastwards across the transect from minimum values of 17 µmol m$^{-2}$ d$^{-1}$ in ST9 to maxima of 295 µmol m$^{-2}$ d$^{-1}$ in ST7 (Fig. 4).

## 4 Discussion

### 4.1 Regional patterns of P cycling

The advent of highly sensitive nanomolar techniques for determining oceanic phosphate has improved our view of phosphate stocks and fluxes in the surface waters of the oligotrophic ocean (Wu et al., 2000; Mather et al., 2008; Letelier et al., 2019). Indeed, traditionally seen as an invariant pool, surface phosphate has recently revealed crucial spatial and temporal patterns with

still poorly known consequences for ocean biogeochemistry (Pulido-Villena et al. 2010; Djaoudi et al., 2018a; Martiny et al., 2019). In this study, surface nanomolar phosphate data provided new insights on the functioning of the phosphorus cycle in upper P-depleted waters of the Mediterranean Sea.



The increasing oligotrophy toward the east is a seminal feature in Mediterranean biogeochemistry (D'ortenzio and d'Alcala, 2009; Mermex group, 2011) alongside the deepening of the phosphacline with longitude (Moutin and Raimbault, 2002; Pujo-Pay et al., 2011, Pasqueron de Fommervault et al., 2015). However, except for a few studies (Moutin et al., 2002, Thingstad et al., 2005, Pulido-Villena et al. 2010; Djaoudi et al., 2018a), phosphate concentration above the phosphacline (i.e. inside the phosphate depleted layer) was set to zero due to methodological limitations, precluding the assessment of vertical and regional variability of this key phosphate pool. Surface nanomolar phosphate data obtained in this study fall within the same order of magnitude than previously reported values for the open Mediterranean Sea during the stratification period by Van Wambeke et al. (1999), Krom et al. (2005), Pulido-Villena et al. (2010) and Djaoudi et al. (2018a). From west to east, the three sub-basins visited in this study exhibited surface phosphate concentrations indicating moderate to severe P-deficiency with values in the Ionian basin (6 nM average) as low as other P-deficient regions such as the North Atlantic subtropical gyre (9 nM, Mather et al., 2008).

One striking feature of the longitudinal pattern in P cycling observed in this study is the difference in the degree of variability among the studied parameters. Indeed, little variability was observed in phosphate concentrations, with average values ranging from 15 nM in the Algerian basin to 6 nM in the Ionian basin. In contrast, alkaline phosphatase (AP) activities varied over one order of magnitude across the longitudinal transect. AP are mostly considered as inducible enzymes and, therefore, AP activity has been used as an index of P-limitation (Hoppe, 2003). This has been particularly examined in the Mediterranean Sea (Sala et al., 2001; Van Wambeke et al., 2002; Thingstad and Mantoura, 2005; Zaccone et al., 2012). In this study, POP-normalized AP activity (AP-POP) exhibited the highest rates in the easternmost station ($0.5$ h$^{-1}$) similar to other P-limited regions (e.g. Sargasso Sea $0.3$ h$^{-1}$, Cotner et al., 1997), indicating an enhanced P-limitation towards the East. This enhanced AP activity was further reflected in an effective drawdown of DOP, indicating that microorganisms residing in surface waters actually utilize DOP as an alternative source of P. Indeed, the turnover time of spiked MUF-P was tightly linked with longitude. Interestingly, none of these P-limitation indexes correlated with phosphate concentration. This implies that the apparent phosphate concentration may not be the definitive criterion for evaluating P limitation, as previously suggested (Tanaka et al., 2006). Concentrations of biologically available orthophosphate [PO4], based on phosphate turnover time from $^{33}$P-phosphate uptake experiments conducted in the Mediterranean Sea during the stratification period, ranged one order of magnitude between 3 nM near the strait of Gibraltar to 0.2 nM in the Ionian Sea (Moutin et al., 2002). Not only the variability was higher than in our study but also reported phosphate concentrations were lower. This may be a seasonal effect since the PEACETIME cruise took place in late spring while the above cited study was conducted at the end of summer, under even more stratified conditions. Nevertheless, it is possible that phosphate concentration, measured through the widely used colorimetric method, does not necessarily reflect the concentration of bioavailable phosphate. Indeed, the substantial difference in APA rates (ten-fold) and turnover times of spiked DOP substrate observed across the transect indicate that microbial P cycling was highly dynamic despite little differences in phosphate concentration, similarly than reported in other P-deficient regions such as the North Pacific sub-tropical gyre (Suzumura et al., 2012).

## 4.2 Vertical variability of phosphate inside the phosphate depleted layer

In addition to the above described longitudinal patterns, the nanomolar phosphate data presented in this study allowed assessing its vertical variability inside the phosphate depleted layer (PDL). As said before, the PDL, between the surface and the top of the phosphacline, has been traditionally seen as a homogeneous layer with undetectable phosphate concentration. Yet, our nanomolar phosphate data revealed the presence of phosphate vertical gradients inside the PDL, ranging between 0.01 and 0.3 nM m$^{-1}$ and decreasing from west to east. In a recent study, Djaoudi et al. (2018a) reported, in the western Mediteranean Sea,



phosphate gradients above the phosphacline between 0.04 and 3.2 nM m⁻¹, higher than observed in the current study. These are, to our knowledge, the first reports on such a vertical variability in phosphate concentration in a P-depleted oceanic region. Similar vertical gradients above the nutricline have been reported for nitrate concentration using highly sensitive measurements

in the oligotrophic South China Sea with values between 0.24 and 0.59 nM N m⁻¹, in the upper range of phosphate gradients reported here. On the contrary, nitrate profiles obtained using highly sensitive measurements during the PEACETIME cruise did not reveal positive vertical gradients but instead homogeneous concentrations or higher concentration at the surface mixed layer compared to the base of the nitrate depleted layer (Van Wambeke et al. 2020).

Including nanomolar measurements in phosphate-density diagrams allowed revealing of two regression lines, one inside the PDL
and the other across the phosphacline, with significantly different slopes (i.e. two distinct values of density gradient of phosphate concentration, $\partial C/\partial \rho$, Fig. 3). This is, to our knowledge, the first report of such a two-slope system in phosphate profiles, revealed thanks to both highly sensitive measurements and high-resolution sampling. In the easternmost station, ION, a three-slope system could be depicted from the phosphate-density diagrams (Fig. 3C): a first null $\partial C/\partial \rho$ in the 0-66 m layer, a second slope in the 66-166 and a third one from 166 until the lower bound of the phosphacline at 400 m. In this study, the phosphacline
was defined as the layer showing a maximum and constant density gradient of phosphate concentration. Therefore, by definition, the PDL spans between the surface and 166 m and, thus, the PDL would be divided in two sub-layers, one upper layer with constant phosphate concentration and the layer underneath showing a first density gradient. An alternative explanation to this three-slope system would consider the 0-66 m as the PDL (low and homogeneous phosphate concentration) and a phosphacline layer showing a nonlinear profile where the phosphate gradient varies locally with density as proposed by Omand and
Mahadevan (2015). This case study at ION highlights the potential of the combination of highly sensitive phosphate measurements at high vertical resolution to characterize the shape of oceanic phosphacline in the upper layers of P-depleted oceanic regions.

Nanomolar phosphate data reported in this study modify the picture of a homogeneous pool of phosphate between the surface and the phosphacline with concentrations theoretically set to zero. Under that traditional view, during the stratification period,
the upper waters of the euphotic zone would not receive any phosphate fluxes from below, since fluxes associated with the phosphacline would only reach the base of the euphotic zone. This study allows picturing an alternative scenario in which diapycnal fluxes of phosphate potentially reach the upper layer of the euphotic zone. In the absence of turbulence measurements during the PEACETIME cruise, phosphate gradients were scaled to flux units by applying a constant value of turbulent kinetic energy dissipation rate (ε) to equation (5) defined in section 2.5. Obviously, the sensitivity of computed diapycnal fluxes to the ε
term in the equation questions these estimates. For comparison, Moutin and Raimbault (2002) considered a ε value of 7 x 10⁻¹⁰ W kg⁻¹ (one order of magnitude lower than in this study) to estimate vertical phosphate fluxes across the phosphacline which ranged between 0.2 and 3.4 μmol P m⁻² d⁻¹. Applying this ε value to the density gradients of phosphate concentration measured in this study across the phosphacline would decrease computed fluxes from 4.5-14.4 μmol m⁻² d⁻¹ to 0.4-1.2 μmol m⁻² d⁻¹. Now that analytical locks have been "opened" allowing to detect consistent density gradients of phosphate above the phosphacline, future
concomitant measurements of microstructure measurements of turbulent dissipation rate or modeled $K_z$-profiles (Costa et al. 2017) will allow quantifying with better accuracy the vertical flux of new P from deeper layers reaching the upper waters of the euphotic zone.

Despite the above described uncertainties of our calculations, our estimates are in the lower range, but in the same order of magnitude, of the only diapycnal fluxes of P inside the PDL reported based on simultaneous turbulence microstructure and high-





resolution chemical measurements in the oligotrophic South China Sea (0.21-0.44 µmol P m$^{-2}$ d$^{-1}$) (Du et al., 2017). We thus
        assume our estimates to be valid enough for the purpose of the following section of this work, i.e. to assess the relative
        contribution of external and internal sources potentially contributing to phosphate supply to the mixed layer.

### 4.3 External sources of P to the upper waters of the Mediterranean Sea

        Together with diapycnal fluxes, the surface mixed layer can receive new P from above, through atmospheric deposition (Pulido-
Villena et al. 2010, Richon et al. 2019), and through lateral advection (Palter et al. 2005). Lateral transport can exceed the
        vertical delivery of nutrients and dominate nutrient budgets in the sub-tropical ocean gyres (Letscher et al. 2016). Mesoscale
        eddies were observed in the study region during the cruise (Guieu et al. 2020). However, there was very little variability in
        surface phosphate concentration precluding the presence of important horizontal gradients. We thus assume phosphate supply
        through lateral advection to be negligible and consider atmospheric deposition and diapycnal fluxes from below as the two main
external sources of phosphate to the phosphate depleted layer.

        Dry deposition of soluble phosphorus exhibited background values across the transect around 0.237 ± 0.140 µmol m$^{-2}$ d$^{-1}$
        (average value for all stations except FAST), on the lower range of previously reported coastal measurements (Markaki et al.
        2003). One Saharan dust event, recorded at FAST (Guieu et al. 2020) was responsible for the highest dry deposition flux, 0.995
        µmol m$^{-2}$ d$^{-1}$. The contribution of dry deposition fluxes to total external fluxes, relative to diapycnal fluxes, increased eastwards,
mainly driven by the decrease in diapycnal fluxes (Fig. 4, Table S1). In the western part of the transect, from ST10 to ST4, both
        fluxes were of the same order of magnitude, except for FAST where atmospheric dry deposition was dominant. This result
        contrasts with the longtime idea that, under stratification conditions, the upper waters of the Mediterranean Sea receive new P
        mainly exclusively from the atmosphere. Although this finding must be taken cautiously given the uncertainties in the estimation
        of diapycnal fluxes, it opens exciting questions on the biogeochemical response of the Mediterranean Sea, and more generally of
marine oligotrophic regions, to expected changes in atmospheric inputs and stratification regimes (Powley et al., 2017). From
        ST5 to the easternmost part of the transect, diapycnal fluxes of phosphate were negligible compared to atmospheric deposition
        (Fig. 4, Table S1). It needs to be remembered here that dry deposition data were derived from measurements of total soluble
        phosphorus including inorganic and organic fractions. Given the potential high contribution of the organic fraction to total
        soluble phosphorus in atmospheric deposition (e.g. Djaoudi et al. 2018b), phosphate fluxes from dry deposition were likely
overestimated. The occurrence of rain events exacerbated the importance of atmospheric deposition in supplying new P to the
        surface layer. This was particularly true at FAST due to the occurrence of a Saharan dust event (Guieu et al. 2020) with an
        estimated dust flux between 38 and 55 mg m$^{-2}$, mainly by wet deposition (Bressac et al., this issue, submitted., Desboeufs et al.
        this issue, in prep).

        The sum of estimated diapycnal fluxes and measured atmospheric deposition would represent an external supply of new
phosphate to the surface layer between 0.11 and 2.19 µmol m$^{-2}$ d$^{-1}$ (Table S1). Assuming a C:P molar ratio of 130 in
        phytoplankton (mean value of sorted phytoplanktonic cells in P-depleted conditions, Martiny et al. 2013), such new phosphate
        would support between 14 and 285 µmol C m$^{-2}$ d$^{-1}$ of new production. Based on primary production (PP) rates reported for the
        PEACETIME cruise (Marañón et al. 2021), PP integrated over the mixed layer would range between 817 and 3611 µmol C m$^{-2}$ d$^{-1}$
        from which 82-361 would be new production based on an average f-ratio of 0.1 previously reported for the Mediterranean Sea
(Powley et al. 2017 and references therein). Under these assumptions, external supply of phosphate to the surface layer would
        support, on average, ca. 30% of new production, except at FAST where external supply would support up to 90 % (Table S1). A
        number of explanations can be proposed for this apparent imbalance related to either overestimation of sinks (new production) or





underestimation of sources. In the first case, applying higher C:P ratios to convert P fluxes into C fluxes would contribute to reducing the imbalance. Due to a great cellular plasticity to local phosphate availability (Moore et al., 2013), particulate C:P

ratios have been shown to be extremely dependent on ambient phosphate concentration with values as high as 200 for phosphate concentrations reaching zero (Galbraith and Martiny, 2015). Applying this ratio to our estimations reduces the imbalance and the contribution of external sources to new production reaches values higher than 80% in stations FAST, ST5, TYR and ION, dominated by atmospheric fluxes (Table S1).

Underestimation of the atmospheric flux may have also contributed to the mismatch between external supply of phosphate and

net productivity across the study transect in the Mediterranean Sea. Only the soluble/dissolved fraction in dry deposition and wet deposition, respectively, has been considered for the estimation. However, along the transect, the percentage of soluble P was highly variable roughly ranging between 10 and 70% of total P (Fu et al. this issue, in prep). Thus, surface waters might have been further enriched in phosphate after dissolution of aerosols in seawater upon deposition (Pulido-Villena et al. 2010).

The imbalance between estimated external sources of P and new production was particularly marked in ST10, the westernmost

station of the transect, where external P supply may explain up to 10% of new production in the best case. It was hypothesized before that lateral transport was a negligible P source all along the cruise transect, mainly based on the regional homogeneity of surface phosphate concentration precluding the presence of important horizontal gradients. ST10 may be an exception to this assumption. Phosphate concentration west off station 10 was not available since this was the westernmost station in the transect precluding the estimation of horizontal gradients. In addition, both measured and altimetry-derived currents indicate the presence

of a mesoscale eddy centered at 1.43°E in the vicinity of ST10 (Fig. 11 in Guieu et al.. 2020). *S*eawater properties of water masses around the station were characteristic of modified Atlantic waters flowing eastward inside the Mediterranean Sea (Guieu et al., 2020). That water mass has been reported to contain significant levels of phosphate (Huertas et al.. 2012) and it is thus plausible that in the western boundaries of the Algerian basin, lateral transport of phosphate notably contributed phosphate external supply to upper waters, explaining the low contribution of both atmospheric deposition and diapycnal fluxes to new

production.

### 4.4 Contribution of regenerated phosphate to total P requirements

Dissolved organic phosphorus represents an alternative source of P for biological activity in P-depleted oceanic regions (Hoppe, 2003, Mather et al., 2008). In this study, the combination of nanomolar phosphate data and alkaline phosphatase (AP) activity showed an enhanced utilization of DOP, particularly in the stations located towards the east of the transect. A variable fraction of

DOP is actually available for hydrolysis by alkaline phosphatase (hereafter AP-DOP). In the North Pacific subtropical gyre, AP-DOP represents 31–48% of total DOP at 20–30 m depth in oligotrophic offshore stations (Suzumura et al. 2012). Previous research in the Mediterranean Sea has reported comparable percentages ($31 \pm 18$ %, Djaoudi et al. 2018b and unpublished data). Applying this percentage to surface DOP concentration measured in this study yields AP-DOP concentrations between 3 and 25 nM. These estimations fall well below the 0.025-1 µM MUF-P range used to compute AP kinetics. We estimated the amount of

phosphate released through enzymatic degradation of DOP by computing *in situ* DOP hydrolysis fluxes for alkaline phosphatase ($AP_{insitu}$). $AP_{insitu}$ was largely below (about 10 times lower) than $V_m$ rates, ranging from 0.07 nmol L$^{-1}$ h$^{-1}$ in ST10 to 0.66 nmol L$^{-1}$ h$^{-1}$ in ST7, in the Ionian basin. These values are slightly higher (although comparable) than those reported for the North Pacific subtropical gyre (0.01-0.07 nmol L$^{-1}$ h$^{-1}$ , Suzumura et al. (2012) and 0.01-0.20 nmol L$^{-1}$ h$^{-1}$, Duhamel et al. (2011)). In contrast, noticeably lower $AP_{insitu}$ has been reported for the North Atlantic subtropical gyre (0.0002 nmol L$^{-1}$ h$^{-1}$, derived from Michaelis

Menten equations and DOP concentration in Mather et al. (2008)). In this study, we estimated Michaelis Menten parameters



based on a multiple-concentration method in which MUF-P was added over a range of concentrations from 25 to 1000 nM. This range is consistent with those used in Suzumura et al. (2012) and Duhamel et al. (2011) studies but markedly lower than in Mather et al. (2008) (1-750 µM). Using MUF-P concentrations much higher than DOP concentration leads to overestimations of Km which, in turn, results in too low $AP_{insitu}$ rates. Generally, enzymatic multiple kinetics are common and using high

fluorogenic substrate concentrations will lead to underestimating in situ hydrolysis rates (Van Wambeke et al., 2021 and references therein).

To scale the estimated $AP_{insitu}$ within the P cycle, we computed the daily fraction of phosphate potentially released by AP activity and the contribution of $AP_{insitu}$ to P demand. Phosphate potentially released by AP activity was calculated as the ratio of $AP_{insitu}$ to phosphate concentration, expressed as a percentage of the ambient phosphate concentration ($AP_{insitu}$/phosphate, % days$^{-1}$). We

found highly variable proportions from 11 % d$^{-1}$ to 100% d$^{-1}$ except for ST07 where the proportion was anomalously higher than 100% d$^{-1}$. Except for this station, these values are in the upper range although comparable to those reported by Duhamel et al (2011) (0.1-63.2 % d$^{-1}$) and much higher than in Suzumura et al. (2012) (1.9-3.3 % d$^{-1}$). This confirms that DOP hydrolysis by alkaline phosphatase activity is a major process supplying phosphate to the surface waters of the Mediterranean Sea. The contribution of $AP_{insitu}$ to P demand was estimated by comparing $AP_{insitu}$ integrated over the mixed layer with total P requirements

(TPR) estimated from primary production and heterotrophic prokaryotic production data reported in Marañon et al. (2021) and Van Wambeke et al. (2021), assuming varying C:P ratios for both phytoplankton and heterotrophic prokaryotes (Table S2, Fig. 5). $AP_{insitu}$ integrated over the mixed layer increased eastwards across the transect with minimum values of 16 µmol m$^{-2}$ d$^{-1}$ in ST10 to maxima of 165 and 225 µmol m$^{-2}$ d$^{-1}$ in stations ION and ST7, respectively (Fig. 4, Table S2). Highest TPR were found in stations ST10, FAST and ST6, averaging 42 µmol m$^{-2}$ d$^{-1}$. Minimum TPR was found in stations ST5, TYR and ION averaging

14 µmol m$^{-2}$ d$^{-1}$ (Table S2). Note that these estimates can vary by almost 100% depending on the C:P ratio assumed for phytoplankton and heterotrophic prokaryotes (see Table S2 for details). The lowest contributions of $AP_{insitu}$ to TPR were found in the western part of the transect, in stations ST10 and ST9. In the eastern part of the transect, from stations TYR to ION, the contribution of $AP_{insitu}$ to TPR largely exceeded 100%. Despite the generally high degree of uncertainty, these comparisons confirm the dominance of internal sources to total phosphate supply to the surface waters of the Mediterranean Sea under

stratified conditions, with little variation of this contribution across the longitudinal transect (Fig. 5).

One intriguing aspect of estimated $AP_{insitu}$ rates concerns the anomalously high contribution of DOP hydrolysis to total phosphate requirements in the surface layer (Table S2). Even considering TPR based on C:P ratios in the upper range of possible values, DOP appears to be over-hydrolyzed compared to P requirements. Similar comparisons between $AP_{insitu}$ rates and P requirements have been previously conducted to estimate the degree of dependence of biological activity on the DOP pool in P-depleted

oceanic regions. In the North Atlantic sub-tropical gyre, 20% (12-30%) of the production was estimated to be supported by the DOP pool (Mather et al. 2008). As said before, unsuitable ranges of MUF-P concentration used to compute kinetic parameters, noticeably higher than DOP concentration, can lead to underestimation of $AP_{insitu}$ likely explaining such a low contribution of the DOP pool. Therefore, the higher contributions of $AP_{insitu}$ to TPR found in this study may be partially explained by the more suitable range of substrate used for hydrolysis measurements. Nevertheless, the observed over-hydrolysis compared to TPR is

still intriguing. Assuming ranges of TPR roughly valid for the purpose of the study, we explore hereafter a number of methodological issues that could explain a potential overestimation of computed $AP_{insitu}$ in this study: i) dealing with AP-DOP concentrations; ii) dealing with the conditions of incubation, and iii) related to the fluorogenic artificial substrate used. Concerning the first point, the proportion of AP-DOP to DOP has been reported to be highly variable in the Mediterranean Sea (Djaoudi et al. 2018b) and AP-DOP/DOP ratios as low as 12 ± 7 % have been reported for the western North Pacific (Hashihama



et al. 2013). In this study, we applied a ratio of 30% to estimate AP-DOP concentration. Considering a AP-DOP/DOP ratio as low as 10% would reduce the $AP_{insitu}$ by one third. Accurate determinations of AP-DOP concentration are thus necessary to evaluate the role of the DOP pool in P-deficient oceanic regions. Regarding the conditions of incubation, AP activity is usually measured in dark conditions and so was in this study, thus excluding the potential effect of UV radiation which might be particularly significant in the Mediterranean Sea where the depth of 1% penetration reaching up to 11 m for UV-A and 26 m for

UV-B (Bertoni et al., 2011). AP activity has been found to be negatively affected by UV radiation (Tank et al. 2005), particularly in the dissolved fraction (Garde et al. 2005, Tank et al. 2005). AP occurs noticeably in the dissolved fraction (Baltar, 2018), including in the study region where the contribution of the < 0.2 μm fraction to the bulk AP activity was on average $60 \pm 34$ % (Van Wambeke et al., 2021). An overestimation of measured AP activity in this study cannot thus be excluded. Finally, MUF-P is the preferred artificial substrate for marine studies but natural AP-DOP is a mixture of compounds with different affinities

(Arnosti et al. 2011). Moreover, if MUF-P is immediately accessible to the alkaline phosphatase, access to natural phospho-monoester binding molecules embedded in large matrixes of polymeric material might require preliminary hydrolysis by other ectoenzymes (e.g. lipase for membrane debris for instance), some of them with lower activities than AP. Overall, further AP measurements in P-depleted oceanic regions would benefit for a better representation of in situ conditions. Despite the above described potential overestimation of AP activity and $AP_{insitu}$, the obtained numbers are distinct enough to confirm the domination

of internal sources in the total phosphate supply to the surface waters of the Mediterranean Sea under stratified conditions, with little variation of this dominance across the longitudinal transect.

## 5 Conclusions

This study presents the first cross-basin dataset in the Mediterranean Sea based on collocated measurements of phosphate pools at the nanomolar level (phosphate, DOP, POP), alkaline phosphatase activities and atmospheric deposition of soluble/dissolved

P. Microbial phosphorus cycling in the surface layer showed a marked longitudinal gradient despite little variability in phosphate concentration. Nanomolar phosphate measured at high vertical resolution inside the phosphate depleted layer revealed vertical gradients between the surface and the phosphacline challenging the traditional view of a homogeneous phosphate deplete layer with phosphate concentration theoretically set to zero. Density gradients of phosphate concentration inside the PDL allowed estimating diapycnal fluxes of phosphate to the mixed layer and comparing them with another external source, atmospheric

fluxes. Under background conditions of dry atmospheric deposition, inputs of phosphate to the surface layer from above and from below contribute roughly equally in the western Mediterranean Sea. The contribution of atmospheric deposition increases eastwards mainly due to a decrease in diapycnal fluxes. On top of this background scenario, pulsed wet deposition events, particularly associated with Saharan dust events, markedly increase the contribution of atmospheric deposition making it the dominant external source of phosphate to the well stratified surface waters of the Mediterranean Sea. These results open exciting

questions on the relative contribution of external sources from below and above to the surface waters of the Mediterranean Sea under changing conditions of atmospheric deposition and stratification regimes. Phosphate supply to the PDL was clearly dominated by internal sources, estimated through in situ enzymatic hydrolysis of the DOP pool by alkaline phosphatase. Internal fluxes of phosphate within the PDL exceeded total phosphate requirements in most visited stations, particularly in the eastern part of the transect, suggesting a potential overestimation of in situ DOP hydrolysis. Taken together, the results obtained in this

study show a highly dynamic phosphorus pool in the upper layer of the euphotic zone, above the phosphacline. This study also highlights the convenience of combining highly sensitive measurements and high-resolution sampling to precisely depict the shape of phosphate profiles in the euphotic zone with still unexplored consequences on P fluxes supplying this crucial layer for



biogeochemical cycles.

**Data availability**

Underlying research data are being used by researcher participants of the PEACETIME campaign to prepare other papers, and therefore data are not publicly accessible at the time of publication. "Biogeochemical dataset collected during the PEACETIME cruise" (Guieu et al., 2020a) will be accessible at https:// www.seanoe.org/data/00645/75747/ once the special issue is completed (all papers should be published by June 2021).

**Author contribution**

CG and KD designed the cruise strategy. KD was responsible for the collection and analysis of atmospheric deposition assisted by FF and ST. KDJ measured nanomolar phosphate onboard. FVW measured alkaline phosphatase activity assisted by SG. VT managed CTD operations onboard. SB, AD, AP and VT contributed to the calculation of diapycnal fluxes. SN managed phosphate sampling and analysis through standrad techniques. TG analysed DOP concentration. EPV supervised the nanomolar phosphate data and managed the analysis of the dataset. EPV wrote the manuscript with contributions from CG, KD, KDJ, FVW,
SB, AD, AP and VT.

**Competing interests**

The authors declare that they have no conflict of interest

**Special issue statement**

This article is part of the special issue 'Atmospheric deposition in the low-nutrient–low-chlorophyll (LNLC) ocean: effects on
marine life today and in the future (ACP/BG inter-journal SI)'. It is not associated with a conference.

**Financial support**

The project leading to this publication received funding from CNRS-INSU, IFREMER, CEA, and Météo-France as part of the programme MISTRALS coordinated by INSU (doi: 10.17600/17000300) and from the European FEDER fund under project no 1166-39417.

**Acknowledgements**

This study is a contribution of the PEACETIME project (http://peacetime-project.org, last access 07/04/2021), a joint initiative of the MERMEX and ChArMEx components. PEACETIME was endorsed as a process study by GEOTRACES and is also a contribution to IMBER and SOLAS international programs. We thank the captain and crew of the RV *Pourquoi Pas?* for their help during the work at sea. We also warmly thank Fabrizio D'Ortenzio for his insightful comments on the first version of the
manuscript.





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


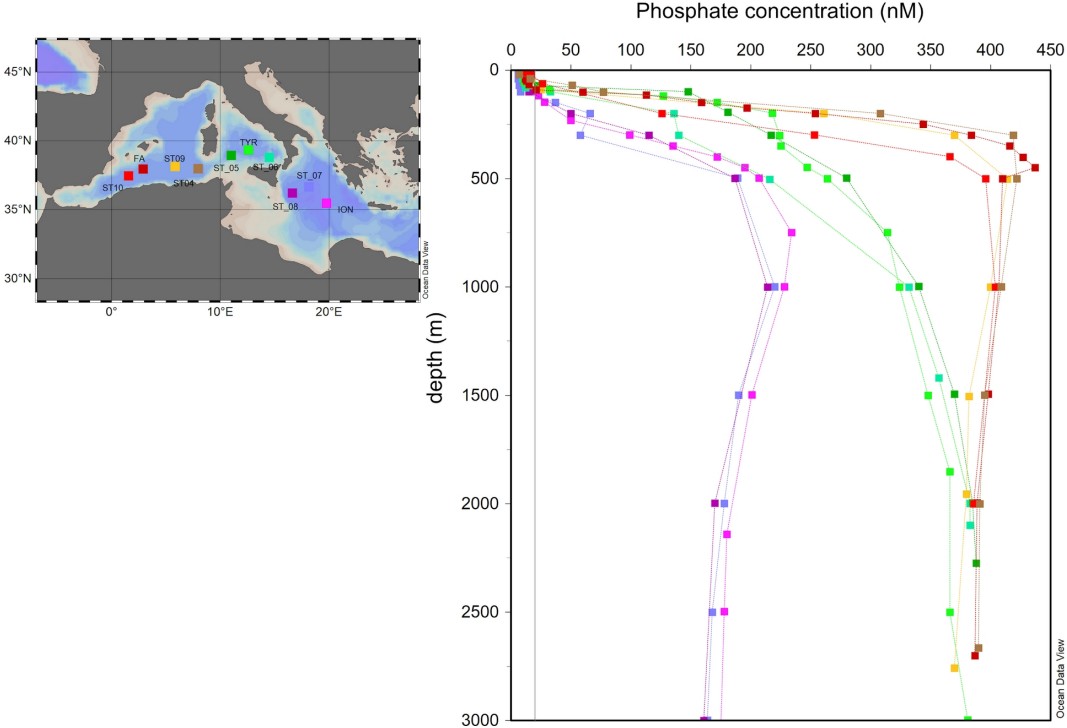

**Figure 1. Station location map and depth profiles of phosphate concentration (nM) between the surface and the bottom in the stations located in the Algerian basin (between 0 and 9°E), in the Tyrrhenian basin (between 10 and 15°E) and in the Ionian basin (beyond 15°E) (Schlitzer, R., Ocean Data View, odv.awi.de, 2021). As described in the Material and Methods section, phosphate vertical profiles were built up from two datasets: one with phosphate concentration obtained through standard technique, with a limit of quantification of 0.02 μM (grey line), and another one with phosphate concentration below 0.02 μM, obtained through the nanomolar LWCC technique.**



Figure 2. Depth of the mixed layer (empty symbols) and the phosphacline (filled symbols, see section 2.5 for details on the calculations)
across the transect (panel A). Longitudinal variability inside the phosphate depleted layer of phosphate concentration (panel B), DOP
concentration (panel C), particulate organic phosphorus (panel D), and AP Michaelis Menten parameters, $V_{max}$ and $K_m$ (panels E and
F). Phosphate and DOP data are represented as box plots (average, standard error and min-max values inside the PDL) as well as raw
data (empty symbols). For POP, $V_{max}$ and $K_m$, data shown were measured at 5 m depth and error bars correspond to measurement
precision (POP) and fitting model standard errors ($V_{max}$ and $K_m$).







**Figure 3. Diagrams of phosphate concentration versus density at FA, TYR and ION from the surface to the bottom and linear**
**regressions between phosphate concentration and density through the phosphacline (black symbols) and inside the PDL (colored**
**symbols). Results of the regression analyses can be found in Table 2 and plots from short stations can be found in Supplementary Fig**
**S1. Vertical grey lines (phosphate concentration = 20 nM) mark the threshold between the nanomolar and micromolar phosphate**
**dataset.**






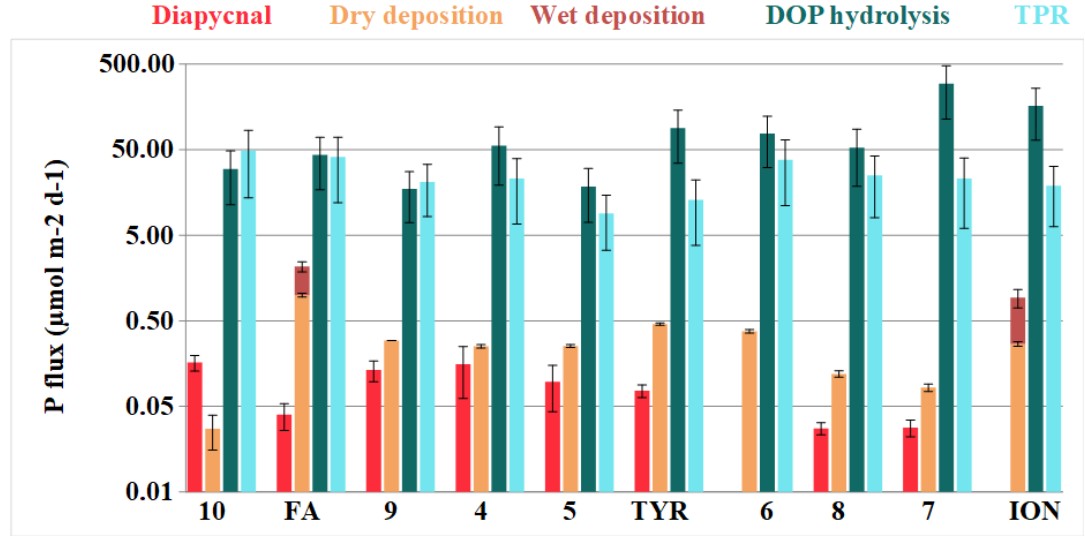

**Figure 4. Phosphate fluxes (µmol m$^{-2}$ d$^{-1}$, logarithmic scale) supplying the mixed layer across the study stations arranged longitudinally: diapycnal fluxes inside the PDL, total (dry plus wet) atmospheric deposition and DOP *in situ* hydrolysis. Estimated total phosphate requirements (TPR) are depicted for comparison (see text and Table S1, S2 for details).**








| Station | Longitude (°E) | MLD (m) | N | $R^2$ | p-value | Slope ($\partial C/\partial \rho$) µmol kg$^{-1}$ | zero-intercept (depletion density) | PD (m) |
|---|---|---|---|---|---|---|---|---|
| ST10 (ALG) | 1.57 | 20 | 5 | 0.76 | 0.0524 | 132.9 ± 42.7 | 27.189 ± 0.392 | 65 |
| FAST (ALG) | 2.92 | 13 | 5 | 0.99 | 0.0003 | 330.5 ± 17.5 | 28.247 ± 0.020 | 87 |
| ST9 (ALG) | 5.84 | 7 | 4 | 0.97 | 0.0132 | 415.3 ± 48.2 | 28.166 ± 0.083 | 90 |
| ST4 (ALG) | 7.98 | 15 | 4 | 0.92 | 0.0392 | 322.3 ± 65.7 | 27.911 ± 0.154 | 74 |
| ST5 (TYR) | 11.02 | 9 | 4 | 0.92 | 0.0406 | 319.5 ± 66.4 | 28.412 ± 0.096 | 61 |
| TYR (TYR) | 12.59 | 9 | 4 | 0.99 | 0.0076 | 578.8 ± 50.8 | 28.608 ± 0.023 | 80 |
| ST6 (TYR) | 14.5 | 18 | 4 | 0.92 | 0.0389 | 291.0 ± 59.1 | 28.427 ± 0.094 | 92 |
| ST8 (ION) | 16.63 | 14 | 3 | 1 | 0.0332 | 802.6 ± 41.9 | 28.910 ± 0.008 | 153 |
| ST7 (ION) | 18.15 | 18 | 5 | 0.89 | 0.0157 | 918.8 ± 185.1 | 28.962 ± 0.027 | 144 |
| ION (ION) | 19.78 | 14 | 4 | 1 | 0.0021 | 935.9 ± 42.6 | 28.928 ± 0.006 | 166 |

**Table 1. Results of the linear fitting computed for each station between phosphate concentration and density at the layer showing the maximum gradient, indicative of the phosphacline. The phosphacline depth (PD) corresponds to the depletion density depth. Note that for ST10, the linear fitting was not significant (p> 0.05) and obtained parameters (slope and zero-intercept) must be taken cautiously.**









| Station | Long (°E) | MLD (m) | PD (m) | Phosphate (nM) | DOP (nM) | POP (nM) | $V_{max}$ (nM h$^{-1}$) | $K_m$ (nM) | $V_{max}$/POP (h$^{-1}$) | $T_{MUF}$ (h) |
|---|---|---|---|---|---|---|---|---|---|---|
| ST10 (ALG) | 1.57 | 18 | 65 | 15 ± 4 | 72 ± 21 | 20 ± 2 | 0.5 ± 0.1 | 123 ± 21 | 0.02 ± 0.00 | 264 ± 57 |
| FAST (ALG) | 2.92 | 13 | 87 | 13 ± 1 | 99 ± 9 | 24 ± 2 | 0.7 ± 0.0 | 138 ± 14 | 0.03 ± 0.00 | 191 ± 22 |
| ST9 (ALG) | 5.84 | 7 | 90 | 13 ± 3 | 56 ± 16 | 19 ± 1 | 2.6 ± 0.1 | 280 ± 28 | 0.14 ± 0.01 | 107 ± 11 |
| ST4 (ALG) | 7.98 | 11 | 74 | 10 ± 5 | 83 ± 24 | 25 ± 2 | 0.6 ± 0.0 | 73 ± 21 | 0.02 ± 0.00 | 128 ± 38 |
| ST5 (TYR) | 11.02 | 15 | 61 | 13 ± 0 | 30 ± 3 | 18 ± 1 | 1.1 ± 0.1 | 112 ± 18 | 0.06 ± 0.01 | 100 ± 16 |
| TYR (TYR) | 12.59 | 9 | 80 | 10 ± 3 | 73 ± 9 | 16 ± 1 | 2.2 ± 0.1 | 96 ± 18 | 0.14 ± 0.01 | 44 ± 9 |
| ST6 (TYR) | 14.50 | 18 | 92 | 11 ± 1 | 35 ± 11 | 17 ± 1 | 2.2 ± 0.1 | 116 ± 10 | 0.13 ± 0.01 | 53 ± 5 |
| ST8 (ION) | 16.63 | 14 | 153 | 13 ± 2 | 31 ± 2 | 14 ± 1 | 1.8 ± 0.1 | 101 ± 28 | 0.13 ± 0.02 | 56 ± 16 |
| ST7 (ION) | 18.15 | 18 | 144 | 6 ± 1 | 56 ± 9 | 13 ± 1 | 3.7 ± 0.1 | 81 ± 11 | 0.28 ± 0.03 | 22 ± 3 |
| ION (ION) | 19.78 | 14 | 166 | 10 ± 0 | 36 ± 10 | 11 ± 1 | 5.6 ± 0.2 | 138 ± 13 | 0.53 ± 0.05 | 25 ± 3 |

**Table 2. Summary of biogeochemical features of the phosphorus pool inside the phosphate depleted layer at each investigated station. See Table 1 and text (section 2.5) for details on the calculation of the phosphacline depth (PD).**







| Station | Longitude (°E) | MLD (m) | PD (m) | N | $R^2$ | p-value | Slope ($\partial C/\partial \rho$) μmol kg$^{-1}$ |
|---|---|---|---|---|---|---|---|
| 10 (ALG) | 1.57 | 20 | 65 | 4 | 0.92 | 0.039 | 10.6 ± 2.2 |
| FA (ALG) | 2.92 | 13 | 87 | 8 | 0.56 | 0.032 | 2.6 ± 0.9 |
| 9 (ALG) | 5.84 | 7 | 90 | 3 | 0.87 | 0.067 | 8.7 ± 2.4 |
| 4 (ALG) | 7.98 | 15 | 74 | 3 | 0.73 | 0.346 | 10.2 ± 6.1 |
| 5 (TYR) | 11.02 | 9 | 61 | 3 | 0.62 | 0.212 | 6.3 ± 3.5 |
| TYR (TYR) | 12.59 | 9 | 80 | 7 | 0.88 | 0.002 | 5.0 ± 0.8 |
| 6 (TYR) | 14.5 | 18 | 92 | 4 | 0.01 | 0.914 | -0.1 ± 1.2 |
| 8 (ION) | 16.63 | 14 | 153 | 5 | 0.95 | 0.025 | 1.8 ± 0.3 |
| 7 (ION) | 18.15 | 18 | 144 | 5 | 0.87 | 0.020 | 1.8 ± 0.4 |
| ION (ION) | 19.78 | 14 | 166 | 7 | 0.07 | 0.582 | 0.2 ± 0.4 |

**Table 3. Results of the linear fitting computed for each station between phosphate concentration and density at the phosphate depleted layer (between the surface and the phosphacline). The slope of the regression line corresponds to the density gradient in phosphate concentration inside the PDL.**







|  | Long | PD | phosphate | DOP | POP | $V_{max}$ | $K_m$ | $V_{max}$/POP | $T_{MUF}$ |
|---|---|---|---|---|---|---|---|---|---|
| Long | 1.0000 | | | | | | | | |
| PD | 0.71 p= 0.009 | 1.0000 | | | | | | | |
| DIP | -0.660 p= 0.038 | -0.259 p= 0.470 | 1.0000 | | | | | | |
| DOP | -0.682 p= 0.030 | -0.396 p= 0.257 | 0.0354 p= 0.923 | 1.0000 | | | | | |
| POP | -0.848 p= 0.002 | -0.757 p= 0.011 | 0.4070 p= 0.243 | 0.721 p= 0.019 | 1.0000 | | | | |
| $V_{max}$ | 0.758 p= 0.011 | 0.780 p= 0.008 | -0.5710 p= 0.085 | -0.470 p= 0.170 | -0.819 p= 0.004 | 1.0000 | | | |
| $K_m$ | -0.334 p= 0.346 | -0.108 p= 0.767 | 0.3773 p= 0.282 | -0.003 p= 0.994 | 0.046 p= 0.899 | 0.164 p= 0.652 | 1.0000 | | |
| $V_{max}$/POP | 0.7581 p= 0.011 | 0.801 p= 0.005 | -0.5438 p= 0.104 | -0.450 p= 0.192 | -0.809 p= 0.005 | 0.983 p= 0.000 | 0.051 p= 0.890 | 1.0000 | |
| $T_{MUF}$ | -0.938 p= 0.000 | -0.510 p= 0.132 | 0.7101 p= 0.021 | 0.619 p= 0.056 | 0.751 p= 0.012 | -0.740 p= 0.014 | 0.130 p= 0.721 | -0.685 p= 0.029 | 1.0000 |

**Table 4. Correlation coefficients among biogeochemical variables related to the phosphorus pool inside the phosphate depleted layer (n= 10).**