# Peer review of "Phosphorus cycling in the upper waters of the Mediterranean Sea (Peacetime cruise): relative contribution of external and internal sources"

_Biogeosciences, 2021_

## Author Comment (AC1)

We would like to thank Anonymous Referee #1 for his/her comments and suggestions on our manuscript. Please find below our responses (**in bold blue**) as well as the associated modifications in the revised version of the manuscript (*in italics*).

Anonymous Referee #1

General comments:

The authors report phosphorus (P) dynamics on the zonal transect in the Mediterranean Sea using a sensitive analytical technique for determining nanomolar P. The technique enabled to represent not only spatial variations in nanomolar P but also their upward flux in the oligotrophic upper water. The authors found a west-east gradient of P dynamics from data on nanomolar P distribution/flux and DOP hydrolysis. Based on these data as well as atmospheric P deposition data, they analyzed P budgets in the mixed layer and found a large external supply of atmospheric deposition relative to upward supply and an importance of internal DOP hydrolysis for sustaining primary production. The budgetary analysis includes some assumptions, but this approach using nanomolar P data is new and is necessary for understating P dynamics in oligotrophic oceans. Overall, the manuscript is well structured and is scientifically implicated. Thus, I consider that this study is suitable for publishing in *Biogeosciences* after addressing specific comments below.

Specific comments:

L24, 25, 27: "L-1 or kg-1"

"-1" should be superscript.

**This will be corrected in the revised manuscript.**

L45: "physical structuration of the water column"

What is "physical structuration"? Is it physical phenomenon such as mixing and/or diffusion? Please clarify.

**We apologize for the awkward phrasing. It will be modified as follows:**

*Concentration of dissolved nutrients are kept low by microorganisms within the euphotic zone and their replenishment is partly driven by physical structure of the water column which constrains vertical nutrient transport (Lewis et al., 1986).*

L72: "North Pacific (Duhamel et al., 2011)"

In addition to Duhamel et al., Sato et al. (2013 BG, doi:10.5194/bg-10-7677-2013) is citable as a representative AP study in the North and South Pacific Ocean.

**The reference will be added.**

L119: "4.5% (n=10)"

What is the concentration level of sample for determining the precision? Please specify the concentration with 4.5%.

**The concentration of the sample used to assess the precision was 11 nmol L$^{-1}$. This information will be added in the revised manuscript.**

*"The analytical precision, estimated through repeated measurements of a surface seawater sample with phosphate concentration of 11 nmol L$^{-1}$, was 4.5% (n= 10)."*

L123: "two 100 nM solutions of glycerol-phosphate and glucose-phosphate"

These compounds are analogs of labile phosphate ester. Did you try to examine the digestion efficiency using a refractory P analog such as phosphonate with C-P bond?

**No, we didn't, although previous studies have reported satisfactory recovery efficiencies of UV-digestion on phosphonate analogs (Armstrong et al. 1966). However, it is true that UV-digestion generally returns lower recoveries than chemistry-based oxidation methods (Karl, 2014 and references therein) and, therefore, DOP estimated through UV-digestion should be considered as a lower limit of ambient DOP concentration. We will add a sentence on this issue in the revised manuscript.**

*"The efficiency of digestion on organic P compounds other than phosphoesters was not assessed in this study, although UV-digestion has also proven to return complete hydrolysis of phosphonate compounds (Armstrong et al. 1966). However, it is true that UV-digestion generally returns lower recoveries than chemistry-based oxidation methods (Karl and Björkman, 2015 and references therein) and, therefore, DOP estimated in this study through UV-digestion should be considered as a lower limit of ambient DOP concentration."*

*Karl, D. M. and Björkman, K. M.: Dynamics of Dissolved Organic Phosphorus, in Biogeochemistry of Marine Dissolved Organic Matter, pp. 233–334, Elsevier., 2015.*

L167-168: "The mixed layer depth…..the profile (1 m)"

Could you please cite the reference for this statement?

**A reference will be added.**

*The mixed layer depth (MLD) was determined at every CTD cast as the depth where the residual mass content (i.e., the vertical integral of the density anomaly relative to surface) was equal to 1 kg m$^{-2}$ (Prieur et al., 2020), with an error of estimation of 0.5 m relative to the vertical resolution of the profile (1 m).*

*Prieur, L., D'Ortenzio, F., Taillandier, V. and Testor, P. (2020), Physical oceanography of the Ligurian sea. In: Migon C., Sciandra A. and Nival P. (eds.), the Mediterranean sea in the era of global change (volume 1), evidence from 30 years of multidisciplinary study of the Ligurian sea. ISTE Sci. Publ. LTD, 49–78. doi:10.1002/9781119706960.ch3*

L198: "unpublished measurements in the study area"

This measurement for determining the fraction of AP-DOP to total DOP is very important, because the result of the AP-DOP fraction significantly influences on in site AP hydrolysis rate. Could you please describe how to measure the AP-DOP here?

**AP-DOP in the mentioned studies (Djaoudi et al. 2017 and unpublished measurements) was measured based on Hashihama et al (2013). Briefly, AP-DOP is calculated as the increase in phosphate concentration after incubation of the seawater sample with a purified alkaline phosphatase. The basis of the measurement will be described in the revised manuscript.**

In situ *alkaline phosphatase hydrolysis rates.* In situ *hydrolysis rates for alkaline phosphatase (AP) were computed using estimated* in situ *concentration of DOP hydrolysable by AP (AP-DOP) as substrate concentration (S) in Michaelis-Menten kinetic equation as following:*

$$V_{insitu} = \frac{\left(V_{max} \cdot AP - DOP\right)}{\left(K_m + AP - DOP\right)}$$

*AP-DOP concentration is usually measured as the increase in phosphate concentration after incubation of a seawater sample with a purified alkaline phosphatase (Hashihama et al. 2013). Therefore it is used as an estimate of phosphate monoesters (Yamaguchi et al. 2019). In the absence of AP-DOP measurements in this study, we applied a fraction of AP-DOP to DOP of 31 ± 18 % (n=36) an average value previously measured in the study region (Djaoudi et al., 2018b, and unpublished data) following the Hashihama et al. (2013) method.*

L247-251: "Using measured density gradient.....14.4 µmol m$^{-2}$ d$^{-1}$."

While this paragraph for phosphate upward flux is informative, I am thinking that the authors can also calculate upward flux of DOP as well as phosphate. If DOP upward supply is significant, it may contribute new production in the mixed layer as well as phosphate upward supply. I am glad the authors consider this point.

**We thank the reviewer for this suggestion. However, DOP profiles inside the phosphate depleted layer did not reveal any positive gradient of concentration with depth suggesting the lack of upward fluxes of DOP. In addition, a previous work on the study region showed a significant decrease of DOP concentration with depth (Djaoudi et al. 2018) inside the phosphate depleted layer suggesting downward rather than upward DOP fluxes.**

L291: "PO4"

"4" should be subscript.

**This will be corrected in the revised manuscript.**

L306: "nM m$^{-1}$"

In the Results section, vertical gradient of phosphate is expressed as a unit of μmol kg$^{-1}$. Why do the authors use "nM m$^{-1}$" here? I think it is better to use same unit throughout the manuscript.

**We have used nM m$^{-1}$ instead of μmol kg$^{-1}$ in this part of the discussion in order to compare our results to phosphate vertical gradients reported by Djaoudi et al (2018a) who expressed them in nM m$^{-1}$. We will slightly rephrased this sentence to avoid confusion between units.**

*Yet, our nanomolar phosphate data revealed the presence of phosphate vertical gradients inside the PDL (except in stations 6 and ION) ranging between 1.8 and 10.6 μmol kg$^{-1}$ and decreasing from west to east. In a recent study in the western Mediterranean Sea, Djaoudi et al. (2018a) reported phosphate gradients over depth above the phosphacline between 0.04 and 3.2 nM m$^{-1}$, higher than observed in the current study (0.01-0.3 nM m$^{-1}$ if expressed as gradient of phosphate concentration over depth).*

L309-313: "Similar vertical gradients…..South China Sea"

The authors need to cite the reference here.

**Following the suggestion below, we will remove this sentence.**

L309-313: "Similar vertical gradients…..(Van Wambeke et al. 2020)"

Why do the authors describe vertical gradient of nitrate here? Since this subsection is for vertical variations in phosphate as indicated by its title, the description of nitrate is likely off-topic.

**We will remove these sentences on the vertical gradients of nitrate, as suggested.**

L339: "analytical locks"

What is it?

**We will replace it by "technical limitations".**

L345: "oligotrophic South China Sea (0.21-0.44 μmol P m$^{-2}$ d$^{-1}$)"

Hydrographic condition is likely different between the Mediterranean Sea and South China Sea. Can you simply compare vertical P fluxes between these different areas?

**This comparison was intended to frame the order of magnitude of our estimated fluxes since we did not have *in situ* measurements of ε. We will slightly modify the sentence to precise that fluxes might indeed be different as a consequence of different hydrological conditions between the two regions. Moreover, we will add a reference to very recent publication (Hashihama et al. 2021) which reports nanomolar phosphate fluxes to the mixed layer of the subtropical North Pacific of the same order of magnitude than our estimates.**

*Despite the above described uncertainties of our calculations, our estimates are in the lower range, but in the same order of magnitude, of the only reported diapycnal fluxes of P reaching the upper euphotic zone based on simultaneous turbulence microstructure and high-resolution chemical measurements in the oligotrophic South China Sea (0.21-0.44 μmol P m-2 d-1) (Du et al., 2017) and in the sub-tropical North Pacific (Hashihama et al. 2021). Thus, despite hydrological differences among these oceanic regions, we ssume our estimates to be valid enough for the purpose of the following section of this*

*work, i.e. to assess the relative contribution of external and internal sources potentially contributing to phosphate supply to the mixed layer.*

*Hashihama, F., Yasuda, I., Kumabe, A., Sato, M., Sasaoka, H., Iida, Y., Shiozaki, T., Saito, H., Kanda, J., Furuya, K., Boyd, P. W. and Ishii, M.: Nanomolar phosphate supply and its recycling drive net community production in the subtropical North Pacific, Nature Communications, 12(1), doi:10.1038/s41467-021-23837-y, 2021.*

L352-353: "However…..horizontal gradients"

The authors need to show the evidence for little horizontal gradient of surface phosphate here. In addition to the west-east gradient, the north-south gradient is likely important for lateral nutrient supply.

**We agree. During the Peacetime cruise, three additional stations were sampled although they were not included in this study (cf map below). The location of these three stations, together with ST04, allows to check surface DIP variability along a north-south gradient. DIP concentration within the phosphate depleted layer in these stations was 11 ± 6 nM in ST01, 10 ± 3 nM in ST02 and 11 ± 6 nM in ST03. These values compare well with ST04 (10 ± 5 nM), at the southernmost latitude of the N-S transect and with the rest of stations across the longitudinal transect (Fig. 2 in the manuscript). This suggests little horizontal gradient of surface phosphate across a north-south gradient in the study area. This information will be added to the revised manuscript.**

[Figure]

*"Together with diapycnal fluxes, the surface mixed layer can receive new P from above, through atmospheric deposition (Pulido-Villena et al. 2010, Richon et al. 2019), and through lateral transport (Letscher et al. 2016). In this study, there was very little variability in surface phosphate concentration among the stations excluding the presence of important horizontal gradients across the longitudinal transect. Moreover, surface phosphate concentration in three stations located further north (between 42°N and 39°N along a north-south gradient) was homogeneous and similar to the study transect (11 ± 6 nM in ST01, 10 ± 3 nM in ST02 and 11 ± 6 nM in ST03, data not shown) suggesting also the absence of horizontal gradients across a north-south gradient. We thus assume that phosphate supply through*

*lateral transport was not dominant during the cruise and consider atmospheric deposition and diapycnal fluxes from below as the two main external sources of phosphate to the phosphate depleted layer.*

L354: "lateral advection to be negligible"

This is a contradiction with the paragraph in L394-405. Please revise.

**We agree that this part of the discussion could appear contradictory since it mixed up mechanisms and processes occurring at different spatio-temporal scales. We have rephrased this section considering two different scales of lateral fluxes: large scale (basin/year) and mesoscale. We will make the distinction in the revised paper between the basin scale (where homogeneity in surface phosphate precludes horizontal gradients) and the particular case of ST10 where lateral phosphate supply may be more important due to local transport specific to the presence of a mesoscale gyre (Guieu et al. 2020 for details).**

*"The imbalance between estimated external sources of P and new production was particularly marked in ST10, the westernmost station of the transect, where external P supply may explain up to 10% of new production in the best case. This contrasts with the rest of stations in the Algerian basin where external P supply explained between 30 and 90 %. As said before, it was hypothesized that, at the basin scale, lateral transport was not a dominant P source mainly based on the homogeneity of surface phosphate concentration excluding the presence of important horizontal gradients. Nevertheless, ST10 might be an exception to this assumption. Using satellite derived currents, we observed the presence of a mesoscale eddy and ST10 was located on the edge of this eddy (Guieu et al. 2020). This can imply a mesoscale transport of nutrients from Atlantic water containing significant levels of phosphate (Huertas et al.. 2012). It may not be excluded that in ST10 lateral transport contributed to phosphate external supply to upper waters, explaining the low contribution of both atmospheric deposition and diapycnal fluxes to new production".*

L400: "Seawater"

"S" should not be italic.

**This will be corrected in the revised manuscript.**

L428: "P demand"

P demand of what?

**We are sorry for the imprecision. We will replace 'P demand' by 'estimated P requirements of phytoplankton and heterotrophic prokaryotes'.**

*To scale the estimated $AP_{insitu}$ within the P cycle, we computed the daily fraction of phosphate potentially released by AP activity and the contribution of $AP_{insitu}$ to total P requirements by both phytoplankton and heterotrophic prokaryotes in the surface mixed layer.*

L442-443: "In the eastern…..exceeded 100%."

The >100% INT/TPR is likely due to preferential recycling of P relative to C as reported in a range of biogeochemical studies (Clark et al. 1998 Nature, doi.org/10.1038/30881; Loh and Bauer 2000 DSR-I, doi:10.1016/S0967-0637(00)00027-3; Paytan et al. 2003 Mar. Chem., doi:10.1016/S0304-4203(03)00052-5; Duhamel et al. 2007 BG, doi:10.5194/bg-4-941-2007; Letscher and Moore 2015 GBC, doi:10.1002/2014GB004904). I recommend the authors use these references to explain the results of >100% INT/TPR.

**We agree with the referee that preferential recycling of P relative to C has been widely reported and that it should be particularly strong under conditions of P-limitation like those likely encountered in the study region. This will be added to the revised manuscript.**

**However, we are not sure that this preferential remineralization would be enough to explain the apparent over-hydrolysis of DOP found in this study. Indeed, the selective removal of phosphorus from DOM presumably reflects the nutrient demand of marine microorganisms which has been considered in this study in the estimation of total P requirements of phytoplankton and heterotrophic prokaryotes. An estimation of resulting C:P ratios will be provided, assuming that P from DOP hydrolysis is fully assimilated by phytoplankton and heterotrophic prokaryotes . C:P ratios would be lower than 20 for heterotrophic prokaryotes and 40 for phytoplankton in all the stations located in the Tyrrhenian and Ionian basins which seems unrealistic even under extreme conditions of P-limitation.**

*One intriguing aspect of estimated AP$_{insitu}$ rates concerns the anomalously high contribution of DOP hydrolysis to total phosphate requirements in the surface layer, exceeding 100% in 7 out of 10 stations (Table S2). Although preferential remineralization of P relative to C is a common feature in oceanic regions (e.g. Clark et al. 1998, Letscher and Moore, 2015), in this study DOP appears to be over-hydrolyzed compared to P requirements. Indeed, if P from DOP hydrolysis was fully assimilated by phytoplankton and heterotrophic prokaryotes, C:P ratios would be lower than 40 and 20, respectively, in all stations of the Tyrrhenian and Ionian basins, which seems unrealistic.*

L458-460: "the proportion of AP-DOP to DOP…..(Hashihama et al. 2013)."

In addition to Djaoudi et al. (2018b) and Hashihama et al. (2013), Yamaguchi et al. (2021 Front. Microb., doi: 10.3389/fmicb.2020.570081) recently reported a review of the proportion of AP-DOP to DOP. Table 1 in Yamaguchi et al. is useful for this study, and the multiple comparison of AP-DOP/DOP ratios in different regions would strengthen the discussion here.

**Thank you for the suggestion. We will add this reference to the revised manuscript to strengthen this part of the discussion.**

*Concerning the first point, the proportion of AP-DOP to DOP has been reported to be extremely variable among oceanic regions (Yamaguchi et al. 2021). In this study, we applied a ratio of 30% to estimate AP-DOP concentration based on data reported by Djaoudi et al. 2018 for the Mediterranean Sea. However, considering a AP-DOP/DOP ratio as low as 10% would reduce the APinsitu by one third.*

L461-462: "Accurate determinations…..P-deficient oceanic regions"

I think the authors generally consider phosphate ester as AP-DOP. However, phosphonate is also utilized by marine diazotroph *Trichodesmium* (Dyhrman et al. 2006 Nature, doi:10.1038/nature04203) and eukaryotic phytoplankton (Whitney and Lomas 2019 Limnol. Oceanogr. Lett., doi: 10.1002/lol2.10100). Therefore, we now should consider both phosphate ester and phosphonate as bioavailable DOP. Could you please discuss phosphonate as well as phosphate ester?

**We apologize for the misunderstanding. We indeed consider AP-DOP as phosphomonoesters since it is measured as the difference in phosphate concentration after hydrolysis with alkaline phosphatase which is a phosphomonoesterase. This said, we are aware that AP-DOP (i.e. phosphomonoesters) does not represent the entire pool of bioavailable DOP. To avoid misunderstanding, two new statements will be added to the revised manuscript:**

**1. In the Material and Methods section, the procedure for measuring AP-DOP will be detailed to make it clear that it correspond to the concentration of phosphomonoesters (see response to comment above)**

**2. In the discussion section, we will now mention that our estimations of DOP hydrolysis (internal sources) might be underestimated since they consider only phosphomonoesters as internal P source while other P compounds such as phosphonates are known to provide alternative P.**

*It should be noted here that DOP hydrolysis in this study was assessed solely through alkaline phosphatase activity and it was therefore restricted to phosphate monoesters. Other organic P compounds such as phosphate diesters or phosphonates are known to provide alternative sources of P (Dyhrman et al. 2006, Whitney and Lomas 2019, Thomsom et al. 2020).*

*Thomson, B., Wenley, J., Lockwood, S., Twigg, I., Currie, K., Herndl, G. J., Hepburn, C. D. and Baltar, F.: Relative Importance of Phosphodiesterase vs. Phosphomonoesterase (Alkaline Phosphatase) Activities for Dissolved Organic Phosphorus Hydrolysis in Epi- and Mesopelagic Waters, Frontiers in Earth Science, 8, doi:10.3389/feart.2020.560893, 2020.*

L505-510: "CC and KD…..and VT"

In the Author contribution, authors' initials KD and KDJ are vague. From the author list in L4-6, I can identify two KD but not KDJ. Please correct.

**We agree although we don't see a straightforward solution for distinguishing these two authors who share the same initials. We will ask for assistance to the editorial office.**

References:

Reference style is not consistent. Please check throughout all references.

**The reference style will be checked throughout.**

Figure 3:

"FA, TYR and ION" in the legend should be "FA (panel A), TYR (panel B), and ION (panel C)".

**This will be corrected in the revised manuscript.**

Tables:

In general, the legends of tables should be appeared just above the tables.

Table 3:

The authors should define significant level such as p<0.05.

**This will be corrected in the revised manuscript.**

Table 4:

Please indicate how to derive the correlation matrix in the text or table legend. Did you use Spearman correlation test or Pearson correlation test?

**We used Pearson correlation test. This will be specified in the table legend.**

*"Table 4. Pearson correlation coefficients among biogeochemical variables related to the phosphorus pool inside the phosphate depleted layer (n= 10). PD = phosphacline depth. DIP: dissolved inorganic phosphate concentration. DOP/ dissolved organic phosphorus. POP: particulate organic phosphorus. Vmax: AP maximum hydrolysis velocity. Km: AP Michaelis Menten halfsaturation constant. Vmax/POP: AP maximum hydrolysis velocity normalized to POP. TMUF: turnover time of spiked DOP substrate, MUF-P. Significant correlations at p< 0.05 are highlighted in bold."*

Table S1:

"Sustained PP" may be wrong. I think "Sustained NP" is correct because this is based on external P flux that corresponds to new production (NP) rather than primary production (PP). Also, "Maranon et al." in the legend should be "Marañón et al.".

**We agree and we will replace sustained PP by sustained NP.**

Table S2:

For "TPR$_{moy}$", please explain the meaning of "$_{moy}$".

**We apologize for the mistake. This will be corrected and explained in the legend.**

---

## Author Comment (AC2)

We would like to thank Maurizio Ribera d'Alcala for their comments and suggestions on our manuscript. Please find below our responses (**in bold blue**) as well as the modifications performed in the revised version of the manuscript (*in italics*).

Referee Report, Manuscript Number: BG-2021-94

Phosphorus cycling in the upper waters of the Mediterranean Sea (Peacetime cruise): relative contribution of external and internal sources by Elvira Pulido-Villena, Karine Desboeufs, Kahina Djaoudi, France Van Wambeke, Stéphanie Barrillon, Andrea Doglioli,  Anne Petrenko, Vincent Taillandier,, Franck Fu, Tiphanie Gaillard, Sophie Guasco, Sandra Nunige, Sylvain Triquet, Cécile Guieu

The paper presents the results of a study on the phosphorus cycle in the surface layer of 10 stations of Mediterranean sea in the framework of a cruise aimed at assessing the importance of atmospheric deposition in driving the functioning of the Mediterranean food web, with specific focus on the planktonic component. Almost (see below) all relevant processes have been considered and quantitatively characterized so that the study may be considered exhaustive. This is possibly the first time that almost all the relevant data/processes have been assembled at different sites of the Mediterranean sea to compare the contribution of different sources of phosphorus to sustain the primary production at the sites. Some fluxes, e.g., the diapycnal ones, suffers for the lack of the proper measuments but this does not at all affect the overall picture that derives from the study.

The main results are: 1. that atmopsheric inputs of phosphorus, even taking out dust deposition events, are in the same order (in the WMED) or are definitely larger (in the EMED) than the fluxes from the phosphocline; 2. there is a slight mismatch between new primary production inferred by direct measurements of C assimilation and that sustainable by the observed phosphorus fluxes (but see below); 3. there is fine 'compartimentalization' in the supposed upper homogeneous layer of phosphorus dynamics, which could be assessed thanks to the high sensitivity analytical methods that were used. The authors also drew some conclusions about possible, existing problems on the methods currently used to assess the different paths of phosphorus cycling in the ocean.

I definitely support the publication of the paper also because its results may contribute to an in depth discussion on the distinction between new vs. recycled production.

My only suggestion is to think about one component of the P-cycle that is not discussed, though it might be part of another paper of the same special issue, which is the fate of the POP. I made a very rough computation. Considering the observed P fluxes to the PDL which might be assimilated (Table S1), if redistributed in the whole layer they would increase POP by several nM d-1 (~6 nM d-1 for the ION station with C:P=130), which is in the order of magnitude of the measured POP standing stock. If my computation is correct and the system is in quasi steady state, meaning that there is no sharp accumulation of P in the PDL, this flux must be balanced by the export. Part of the POP, as well as DOP, might be exported below the PDL by sinking particles, though their sinking velocity is not very large considering the prevalence of pico-plankton, by the diapycnal export due to the mixing (DOP and

POP gradients should be opposite to that of Phosphate) and by POP removal by consumption. Flow to the adult metazoans would be missed in the particulate but part of consumption might occur via protozoan grazing and mixotrophy by autotrophic organisms, which would add another term to the P turnover, in addition to that of DOP. This might be considered in the discussion.

**We are not sure to have understood this suggestion and we apologize for this. If we understood well, external fluxes of P to the PDL (0.913 µmol P m$^{-2}$ d$^{-1}$ in the case of ION station), if redistributed in the whole layer (166 m depth), would result in an increase of POP of 0.006 µmol m$^{-3}$ d$^{-1}$, that is 0.006 nM d$^{-1}$ which is two orders of magnitude lower than measured POP (10 nM). If this computation is correct (unless we misunderstood your comment), there would not be a significant net accumulation of POP in the surface layer and the system would not be imbalanced. Nevertheless, we will be happy to discuss this again if we did not get your point.**

Below some minor suggestions/remarks

l.22 "collocated" better co-located

**This will be corrected in the revised manuscript.**

l.27-29 and l.177 I think that "..gradient of phosphate concentration over density" would be more clear than "density gradient"

**This will be corrected in the revised manuscript.**

l.316-317 "was set to zero" ,l.347-348 "has been traditionally seen as a homogeneous layer" and l.372-373 "...the picture of a homogeneous pool of phosphate between the surface and the phosphacline with concentrations theoretically set to zero" I would rephrase them all to convey the concept that the real phosphate concentration and its gradient could not be properly assessed since reported concentrations in the literature were in the range of the detection limit of the classical methodology, which is different from the assumption that were zero. There are plenty of papers and two databases that report measured phosphate concentrations larger then zero in the Mediterranean surface layer.

**We agree and we will rephrase this all throughout the revised version of the manuscript.**

l.387-388 "..our estimates are in the lower range" why in the lower range? even with a lower epsilon they are larger.

**Here we compare diapycnal fluxes of P to the base of the ML (that is, above the phosphacline) between our study (0-0.19 µmol m$^{-2}$ d$^{-1}$) and Du et al (2017) (0.21-0.44 µmol m$^{-2}$ d$^{-1}$). These fluxes are certainly lower than those associated to the phosphacline which should not reach the upper euphotic layer.**

**We realize that the sensitivity test to epsilon performed in the paragraph just above (on the P fluxes across the phosphacline) can lead to confusion and does not add that much to the discussion. We will thus remove it to avoid misunderstanding.**

*Obviously, the sensitivity of computed diapycnal fluxes to the ε term in the equation may question these estimates. Now that analytical locks have been "opened" allowing to detect consistent gradients of phosphate concentration over density above the phosphacline, future concomitant measurements of microstructure measurements of turbulent dissipation rate or modeled Kz-profiles (Costa et al. 2017) will allow quantifying with better accuracy the vertical flux of new P reaching the upper waters of the euphotic zone.*

l.408-409 "This result contrasts with the longtime idea that, under stratification conditions, the upper waters of the Mediterranean Sea receive new P mainly exclusively from the atmosphere" indeed this is what the study shows for the EMED.

**You are right. We have rephrased this sentence to clarify our point which is that atmospheric deposition is not the only external source of P to the mixed layer, particularly in the WMED.**

*"In the western part of the transect, from ST10 to ST4, both fluxes were of the same order of magnitude, except for FAST where atmospheric deposition was dominant. This contrasts with the longtime idea that, under stratification conditions, the upper waters of the Mediterranean Sea receive new P exclusively from the atmosphere. Contrarily, from ST5 to the easternmost part of the transect, diapycnal fluxes of phosphate were negligible compared to atmospheric deposition (Fig. 4, Table S1). Although these findings must be taken cautiously given the uncertainties in the estimation of diapycnal fluxes, it opens exciting questions on the biogeochemical response of the Mediterranean Sea, and more generally of marine oligotrophic regions, to expected changes in atmospheric inputs and stratification regimes (Powley et al., 2017)".*

l.449-453 "..lateral transport of phosphate notably contributed phosphate external supply to upper waters, explaining the low contribution of both atmospheric deposition and diapycnal fluxes to new production" I am a little perplex about this statement because the spatio-temporal scales of lateral transport are different from those assessed with the observations discussed in the paper.

**We agree that this part of the discussion was very confusing since it mixed up mechanisms and processes occurring at different spatio-temporal scales. We have rephrased this section considering two different scales of lateral fluxes: large scale (basin/year) and mesoscale. We will make the distinction in the revised paper between the basin scale (where homogeneity in surface phosphate precludes horizontal gradients) and the particular case of ST10 where lateral phosphate supply may be more important due to local transport specific to the presence of a mesoscale gyre (Guieu et al. 2020 for details).**

*The imbalance between estimated external sources of P and new production was particularly marked in ST10, the westernmost station of the transect, where external P supply may explain up to 10% of new*

*production in the best case. This contrasts with the rest of stations in the Algerian basin where external P supply explained between 30 and 90 %. As said before, it was hypothesized that, at the basin scale, lateral transport was not a dominant P source mainly based on the homogeneity of surface phosphate concentration excluding the presence of important horizontal gradients. Nevertheless, ST10 might be an exception to this assumption. Using satellite derived currents, we observed the presence of a mesoscale eddy and ST10 was located on the edge of this eddy (Guieu et al. 2020). This can imply a mesoscale transport of nutrients from Atlantic water containing significant levels of phosphate (Huertas et al.. 2012). It may not be excluded that in ST10 lateral transport contributed to phosphate external supply to upper waters, explaining the low contribution of both atmospheric deposition and diapycnal fluxes to new production.*

l.521-523 "...the obtained numbers are distinct enough to confirm the domination of internal sources in the total phosphate supply to the surface waters of the Mediterranean Sea under stratified conditions, with little variation of this dominance across the longitudinal transect." This might deserve some more comments (see also above). On one hand this confirms that in the sampled region in late spring recycled production dominates, which was more or less known, but also that phosphorus turnover time is very fast, and that, I suspect, most of the DOP that sustains the recycled production in the EMED dreives from the atmopsheric deposition more than from Atlantic water. What comes from diapycnal fluxes is just recycling on a longer time scale.

**We agree. However, given the uncertainties associated to our results and our 'snapshot' approach, we prefer not to throw any conclusion concerning larger spatio-temporal scales which would not be supported by our data. The fate of atmospheric deposition in the euphotic zone was assessed through a high frequency study of a rain event at the FAST station in Van Wambeke et al. (2021). This work showed a very rapid response of the microbial food web and a return to initial biogeochemical conditions within two days.**

**Van Wambeke, F., Taillandier, V., Deboeufs, K., Pulido-Villena, E., Dinasquet, J., Engel, A., Marañón, E., Ridame, C., and Guieu, C.: Influence of atmospheric deposition on biogeochemical cycles in an oligotrophic ocean system, Biogeosciences Discuss. [preprint], https://doi.org/10.5194/bg-2020-411, in review, 2020.**

References

l.159 2020, Desboeufs et al. also in l.219, l.295, l.418 and Fu et al., also on l.211, l.439 in preparation if it is for Atm. Chem. Phys. is not in this special issue

**We apologize for this. Desboeufs et al. is submitted and in progress of edition for discussion in ACP in this special issue. For Fu et al., it is in prep. and should be submitted before the end of the summer in ACP out of the Special Issue.**

l.214 (Izquierdo et al., 2012) missing in the reference list

**The reference will be added to the list.**

l.318 Van Wambeke et al. (1999) missing

**The reference will be added to the list.**

in the reference list but never cited:

Hashihama, F., Kinouchi, S., Suwa, S., Suzumura, M., and Kanda, J.: Sensitive determination of enzymatically labile dissolved organic phosphorus and its vertical profiles in the oligotrophic western North Pacific and East China Sea, J Oceanogr, 69, 357–367, DOI 10.1007/s10872-013-0178-4, 2013.

**This reference is now cited in the revised manuscript following the suggestion of referee 1.**

Markaki, Z., Loÿe-Pilot, M., Violaki, K., Benyahya, L. and Mihalopoulos, N.: Variability of atmospheric deposition of dissolved nitrogen and phosphorus in the Mediterranean and possible link to the anomalous seawater N/P ratio, Marine Chemistry, 120(1-4), 187–194, 2010.

**Removed**

Table 4 what does it mean Long in a correlation table? some correlations are not particularly useful

**We agree and we will remove non relevant correlations in the revised manuscript.**

| | $PD$ | $DIP$ | $DOP$ | $POP$ | $V_{max}$ | $K_m$ | $V_{max}/POP$ | $T_{MUF}$ |
|---|---|---|---|---|---|---|---|---|
| $PD$ | *1.0000* | | | | | | | |
| $DIP$ | -0.259
 p= 0.470 | *1.0000* | | | | | | |
| $DOP$ | -0.396
 p= 0.257 | 0.0354
 p= 0.923 | *1.0000* | | | | | |
| $POP$ | **-0.757**
 **p= 0.011** | 0.4070
 p= 0.243 | **0.721**
 **p= 0.019** | *1.0000* | | | | |
| $V_{max}$ | 0.780
 **p= 0.008** | -0.5710
 p= 0.085 | -0.470
 p= 0.170 | **-0.819**
 **p= 0.004** | *1.0000* | | | |
| $K_m$ | -0.108
 p= 0.767 | 0.3773
 p= 0.282 | -0.003
 p= 0.994 | 0.046
 p= 0.899 | 0.164
 p= 0.652 | *1.0000* | | |
| $V_{max}/POP$ | 0.801
 **p= 0.005** | -0.5438
 p= 0.104 | -0.450
 p= 0.192 | **-0.809**
 **p= 0.005** | **0.983**
 **p= 0.000** | 0.051
 p= 0.890 | *1.0000* | |
| $T_{MUF}$ | -0.510
 p= 0.132 | **0.7101**
 **p= 0.021** | 0.619
 p= 0.056 | **0.751**
 **p= 0.012** | **-0.740**
 **p= 0.014** | 0.130
 p= 0.721 | **-0.685**
 **p= 0.029** | *1.0000* |

*Table 4. Pearson correlation coefficients among biogeochemical variables related to the phosphorus pool inside the phosphate depleted layer (n= 10). PD = phosphacline depth. DIP: dissolved inorganic phosphate concentration. DOP/ dissolved organic phosphorus. POP: particulate organic phosphorus. Vmax: AP maximum hydrolysis velocity. Km: AP Michaelis Menten half saturation constant. Vmax/POP: AP maximum hydrolysis velocity normalized to POP. TMUF: turnover time of spiked DOP substrate, MUF-P. Significant correlations at p< 0.05 are highlighted in bold.*

Table S2 better to replace Tmoy(enne) with Tavg or T mean

**We apologize for the mistake. This will be corrected.**

---

## Author Response (AR1)

Dear Editor,

Please find attached a revised version of our manuscript including all the revisions proposed in our responses to the comments of both referees.

Best regards,
Elvira Pulido-Villena

---

## Editor Decision (ED1)

[revised manuscript text omitted]
| FAST (ALG) | 2.92 | 13 | 87 | $13 \pm 1$ | $99 \pm 9$ | $24 \pm 2$ | $0.7 \pm 0.0$ | $138 \pm 14$ | $0.03 \pm 0.00$ | $191 \pm 22$ |
| ST9 (ALG) | 5.84 | 7 | 90 | $13 \pm 3$ | $56 \pm 16$ | $19 \pm 1$ | $2.6 \pm 0.1$ | $280 \pm 28$ | $0.14 \pm 0.01$ | $107 \pm 11$ |
| ST4 (ALG) | 7.98 | 11 | 74 | $10 \pm 5$ | $83 \pm 24$ | $25 \pm 2$ | $0.6 \pm 0.0$ | $73 \pm 21$ | $0.02 \pm 0.00$ | $128 \pm 38$ |
| ST5 (TYR) | 11.02 | 15 | 61 | $13 \pm 0$ | $30 \pm 3$ | $18 \pm 1$ | $1.1 \pm 0.1$ | $112 \pm 18$ | $0.06 \pm 0.01$ | $100 \pm 16$ |
| TYR (TYR) | 12.59 | 9 | 80 | $10 \pm 3$ | $73 \pm 9$ | $16 \pm 1$ | $2.2 \pm 0.1$ | $96 \pm 18$ | $0.14 \pm 0.01$ | $44 \pm 9$ |
| ST6 (TYR) | 14.50 | 18 | 92 | $11 \pm 1$ | $35 \pm 11$ | $17 \pm 1$ | $2.2 \pm 0.1$ | $116 \pm 10$ | $0.13 \pm 0.01$ | $53 \pm 5$ |
| ST8 (ION) | 16.63 | 14 | 153 | $13 \pm 2$ | $31 \pm 2$ | $14 \pm 1$ | $1.8 \pm 0.1$ | $101 \pm 28$ | $0.13 \pm 0.02$ | $56 \pm 16$ |
| ST7 (ION) | 18.15 | 18 | 144 | $6 \pm 1$ | $56 \pm 9$ | $13 \pm 1$ | $3.7 \pm 0.1$ | $81 \pm 11$ | $0.28 \pm 0.03$ | $22 \pm 3$ |
| ION (ION) | 19.78 | 14 | 166 | $10 \pm 0$ | $36 \pm 10$ | $11 \pm 1$ | $5.6 \pm 0.2$ | $138 \pm 13$ | $0.53 \pm 0.05$ | $25 \pm 3$ |

**Table 2. Summary of biogeochemical features of the phosphorus pool inside the phosphate depleted layer at each investigated station. See Table 1 and text (section 2.5) for details on the calculation of the phosphacline depth (PD).**

| Station | Longitude (°E) | MLD (m) | PD (m) | N | $R^2$ | p-value | Slope ($\partial C/\partial \rho$) μmol kg$^{-1}$ |
|---|---|---|---|---|---|---|---|
| 10 (ALG) | 1.57 | 20 | 65 | 4 | 0.92 | 0.039 | $10.6 \pm 2.2$ |
| FA (ALG) | 2.92 | 13 | 87 | 8 | 0.56 | 0.032 | $2.6 \pm 0.9$ |
| 9 (ALG) | 5.84 | 7 | 90 | 3 | 0.87 | 0.067 | $8.7 \pm 2.4$ |
| 4 (ALG) | 7.98 | 15 | 74 | 3 | 0.73 | 0.346 | $10.2 \pm 6.1$ |
| 5 (TYR) | 11.02 | 9 | 61 | 3 | 0.62 | 0.212 | $6.3 \pm 3.5$ |
| TYR (TYR) | 12.59 | 9 | 80 | 7 | 0.88 | 0.002 | $5.0 \pm 0.8$ |
| 6 (TYR) | 14.5 | 18 | 92 | 4 | 0.01 | 0.914 | $-0.1 \pm 1.2$ |
| 8 (ION) | 16.63 | 14 | 153 | 5 | 0.95 | 0.025 | $1.8 \pm 0.3$ |
| 7 (ION) | 18.15 | 18 | 144 | 5 | 0.87 | 0.020 | $1.8 \pm 0.4$ |
| ION (ION) | 19.78 | 14 | 166 | 7 | 0.07 | 0.582 | $0.2 \pm 0.4$ |

**Table 3. Results of the linear fitting computed for each station between phosphate concentration and density at the phosphate depleted layer (between the surface and the phosphacline). The slope of the regression line corresponds to the gradient in phosphate concentration over density inside the PDL.**

| | PD | DIP | DOP | POP | $V_{max}$ | $K_m$ | $V_{max}$/POP | $T_{MUF}$ |
|---|---|---|---|---|---|---|---|---|
| **PD** | 1.0000 | | | | | | | |
| **DIP** | -0.259
p= 0.470 | 1.0000 | | | | | | |
| **DOP** | -0.396
p= 0.257 | 0.0354
p= 0.923 | 1.0000 | | | | | |
| **POP** | **-0.757**
**p= 0.011** | 0.4070
p= 0.243 | **0.721**
**p= 0.019** | 1.0000 | | | | |
| **$V_{max}$** | 0.780
**p= 0.008** | -0.5710
p= 0.085 | -0.470
p= 0.170 | **-0.819**
**p= 0.004** | 1.0000 | | | |
| **$K_m$** | -0.108
p= 0.767 | 0.3773
p= 0.282 | -0.003
p= 0.994 | 0.046
p= 0.899 | 0.164
p= 0.652 | 1.0000 | | |
| **$V_{max}$/POP** | 0.801
**p= 0.005** | -0.5438
p= 0.104 | -0.450
p= 0.192 | **-0.809**
**p= 0.005** | **0.983**
**p= 0.000** | 0.051
p= 0.890 | 1.0000 | |
| **$T_{MUF}$** | -0.510
p= 0.132 | **0.7101**
**p= 0.021** | 0.619
p= 0.056 | **0.751**
**p= 0.012** | **-0.740**
**p= 0.014** | 0.130
p= 0.721 | **-0.685**
**p= 0.029** | 1.0000 |

**Table 4. Pearson correlation coefficients among biogeochemical variables related to the phosphorus pool inside the phosphate depleted layer (n= 10). PD = phosphacline depth. DIP: dissolved inorganic phosphate concentration. DOP: dissolved organic phosphorus. POP: particulate organic phosphorus. Vmax: AP maximum hydrolysis velocity. Km: AP**
**Michaelis Menten  constant. Vmax/POP: AP maximum hydrolysis velocity normalized to POP. TMUF: turnover time of spiked DOP substrate, MUF-P. Significant correlations at p< 0.05 are highlighted in bold.**

---

## Author Response (AR2)

Dear Christine Klass,

Thank you for your thorough reading of our manuscript and your constructive comments. We have incorporated most of your remarks and/or changes to the revised version.

Kind regards,
Elvira Pulido-Villena

---

## Author Response (AR3)

Dear Christine Klass,

We have modified the odd sentence in the abstract from "mostly exclusively" to "almost exclusively" in order to keep the sense of the statement.

Kind regards,
Elvira Pulido-Villena

---

## Editor Decision (ED3)

**Phosphorus cycling in the upper waters of the Mediterranean Sea (Peacetime cruise): relative contribution of external and internal sources**

Elvira Pulido-Villena1, Karine Desboeufs2, Kahina Djaoudi1a, France Van Wambeke1, Stéphanie Barrillon1,

5 Andrea Doglioli1, Anne Petrenko1, Vincent Taillandier3, Franck Fu1b, Tiphanie Gaillard1, Sophie Guasco1, Sandra Nunige1, Sylvain Triquet2, Cécile Guieu3

[revised manuscript text omitted]